# BETTER LLM REASONING VIA DUAL-PLAY

## ABSTRACT

Large Language Models (LLMs) have achieved remarkable progress through Reinforcement Learning with Verifiable Rewards (RLVR), yet still rely heavily on external supervision (e.g., curated labels). Adversarial learning, particularly through self-play, offers a promising alternative that enables models to iteratively learn from themselves—thus reducing reliance on external supervision. Dual-play extends adversarial learning by assigning specialized roles to two models and training them against each other, fostering sustained competition and mutual evolution. Despite its promise, adapting dual-play training to LLMs remains limited, largely due to their susceptibility to reward hacking and training instability. In this paper, we introduce PasoDoble, a novel LLM dual-play framework. PasoDoble adversarially trains two models initialized from the same base model: a Proposer, which generates challenging questions with ground-truth answers, and a Solver, which attempts to solve them. We enrich the Proposer with knowledge from a pre-training dataset to ensure the questions' quality and diversity. To avoid reward hacking, the Proposer is rewarded for producing only valid questions that push the Solver's limit, while the Solver is rewarded for solving them correctly, and both are updated jointly. To further enhance training stability, we introduce an optional offline paradigm that decouples Proposer and Solver updates, alternately updating each for several steps while holding the other fixed. Notably, PasoDoble operates without supervision during training. Experimental results show that PasoDoble can improve the math reasoning performance of LLMs.

## 1 INTRODUCTION

Large Language Models (LLMs) have recently made striking progress with Reinforcement Learning using Verifiable Rewards (RLVR) (Shao et al., 2024b; DeepSeek-AI, 2025). In RLVR, a response is rewarded based on whether its final answer matches a ground truth according to verifiable rules. RLVR is simple and robust against reward hacking, and has been used to empower LLMs across various tasks and domains. Applying RLVR at scale, however, still hinges on the availability of large volumes of externally supervised data (i.e., expertly curated task-specific examples, each paired with a correct outcome label), creating a critical bottleneck to progress for tasks and domains for which supervised data are scarce or non-existant (Zhao et al., 2025; Su et al., 2025).

In recent work, several alternatives have been explored to reduce reliance on external supervision in reinforcement learning. Unsupervised reinforcement learning, for instance, derives reward signals directly from the model's own outputs, calculating sequence-level confidence scores (Li et al., 2025; Prabhudesai et al., 2025; Huang et al., 2025a) or relying on output entropy (Agarwal et al., 2025; Cheng et al., 2025). These approaches, however, generally require access to curated task-specific examples for training. Adversarial learning (AL), particularly self-play (Zhao et al., 2025; Cheng et al., 2024; Chen et al., 2025a; Kuba et al., 2025), has also emerged as a promising approach to reduce reliance on external supervision signals. In this paradigm, a model generates its own labelled examples and improves through iterative self-challenging and self-feedback loops. Related *dual-play adversarial training* approaches (e.g., generative adversarial networks (GANs) (Goodfellow et al., 2014)) offer a possibly even more compelling training paradigm by training **two** models adversarially: one model is dedicated to generating challenging tasks or adversarial examples, while the other focuses on solving them. The hope is that the specialized role of each model fosters sustained competition and mutual evolution between the two models.

Research on applying dual-play to LLM training, however, remains limited. One very recent attempt is R-Zero (Huang et al., 2025b), which trains two LLMs *separately* to improve its reasoning capabilities and does so without the need for labelled training data. While the approach is promising, R-Zero trains two LLMs separately rather than adversarially, causing improvements to plateau after only a few iterations (i.e., three). Thus, designing a dual-play framework for training LLMs that enables sustained improvement across training iterations without requiring large amounts of supervised training data remains an open challenge.

In this paper, we propose a novel LLM dual-play framework, PasoDoble, as a potential solution. In contrast to R-Zero, PasoDoble *adverarially* trains two LLMs initialized from the same base model: a Proposer model (self-)generates well-formed challenging questions together with their ground-truth answers, and a Solver attempts to solve each question multiple times. Accordingly, we design the Proposer's reward to be inversely correlated with the Solver's accuracy: the harder the questions (i.e., the lower the Solver's accuracy), the higher the Proposer's reward. Conversely, the Solver is rewarded for solving the questions correctly. Thus, as training iterations proceed, the Proposer strives to generate question-answer pairs that continue to challenge the capabilities of the Solver.

We purposely design PasoDoble to avoid some anticipated problems. First, without sufficient knowledge of the target task/domain, question-answer pair generation by the Proposer is likely to be inadequate. Consequently, PasoDoble presumes access to a high-quality external pre-training knowledge base that serves to ground the Proposer in accurate, diverse, and relevant content for the target task/domain. More so than curated questions or question-answer pairs, we expect background knowledge for most tasks/domains of interest to be readily available. To prevent reward hacking by the Proposer (e.g., creating questions with incorrect answers), PasoDoble rewards it only for *valid* questions—questions for which the Solver's average accuracy across multiple attempts exceeds a predefined threshold (e.g., 20%) and whose diversity score relative to previously generated questions surpasses a set cutoff. At each iteration, we jointly update the Proposer and the Solver. Finally, to mitigate the potential instabilities introduced by adversarial training, we explore an *offline* variation of the PasoDoble training paradigm in which updates to the Proposer and the Solver are decoupled: in each iteration, the Solver is first frozen while the Proposer is trained for several steps to generate questions, from which valid ones are collected. These collected questions are then used to train the Solver for several steps.

We evaluate PasoDoble on math benchmarks (AIME, AMC, GSM8K, MATH-500, and Olympiad-Bench) and show that both its online and offline variants improve performance across diverse base LLMs, including Qwen2.5-0.5B/1.5B/3B-Base and Qwen3-0.6B/1.7B/4B-Base. For instance, the average performance of Qwen3-1.7B-Base improves by **11.42** and **12.75** points (average pass@1 accuracy) under the online and offline settings, respectively. Larger models show more performance gains. For Qwen3-4B-Base, we see an increase by **15.96** and **15.10** points in average accuracy. Importantly, PasoDoble is able to sustain improvements during training in both the online and offline setting for hundreds of update steps, showing a much stronger scaling capacity than R-Zero. Our ablation studies show that joint training of the Proposer and Solver is essential across all models, whereas external knowledge benefits only the larger 1.5B/1.7B/3B/4B models. Overall, our findings demonstrate that PasoDoble not only enables a dual-play framework where two LLMs co-evolve without supervision, but also serves as a bridge between pre-training and post-training by leveraging pre-training knowledge to enhance post-training.

## 2 RELATED WORK

Reinforcement learning with Verifiable Rewards (RLVR) provides outcome-based feedback without requiring supervision over intermediate reasoning steps, thereby mitigating reward hacking. A major bottleneck of RLVR is its reliance on external supervision, i.e., expertly curated tasks and their outcome labels. The goal of this paper is to address this bottleneck.

**Label-free approaches.** Prior work has sought to reduce the supervision burden by developing "label-free" training methods that eliminate the need for ground-truth answers by leveraging confidence maximization (entropy minimization) (Li et al., 2025; Prabhudesai et al., 2025; Agarwal et al., 2025) or calibration (Huang et al., 2025a). These approaches still incur the cost of curating tasks

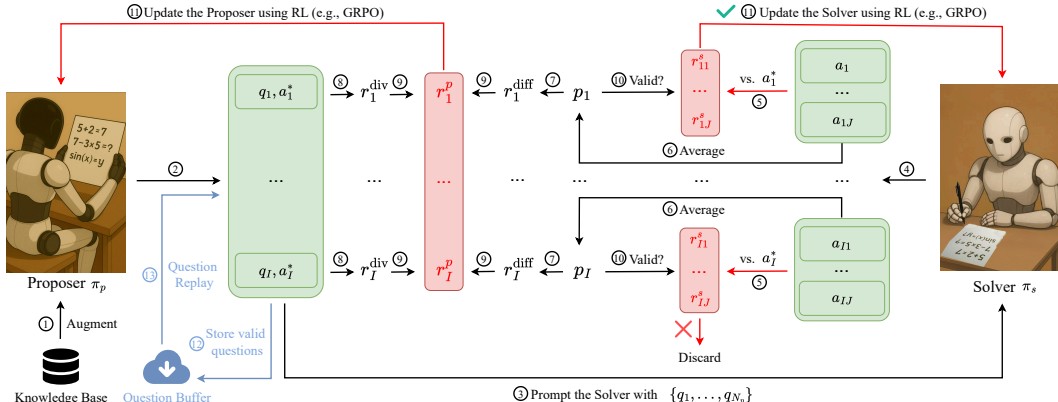

Figure 1: The PasoDoble framework.

(i.e., training examples). Our approach requires no labels and minimizes question curation costs throught the use of readily available texts from the target task/domain.

**Adversarial Learning (AL) approaches.** Adversarial learning (AL) has emerged as a promising direction for removing the need for external supervision, with self-play as a central paradigm (e.g.,AlphaZero (Silver et al., 2018)). The approach has recently been extended to LLMs: Fang et al. (2025) and Chen et al. (2025b) train models to generate and solve increasingly difficult math questions using majority-voted labels. AbsoluteZero (Zhao et al., 2025) extends this idea to coding tasks. A more advanced variant of adversarial learning adopts a dual-model setup, where two models are trained concurrently. The classical example is GANs (Goodfellow et al., 2014), in which a generator and a discriminator compete to produce increasingly realistic outputs. Dual-play has been widely applied to robotics (Pinto et al., 2017), sequence modeling (Yu et al., 2017), and pre-LLM NLP tasks (Chen et al., 2018; Zhang et al., 2017; Nie et al., 2019). More recently, R-Zero (Huang et al., 2025b) trains two LLMs to enhance reasoning, but their training is decoupled rather than adversarial, leading to rapid performance plateauing and even degradation after only three iterations.

## 3 PASODOBLE

In this section, we first present an overview of our PasoDoble framework (§ 3.1), followed by detailed descriptions of the knowledge base (§ 3.2), the Proposer (§ 3.3), and the Solver (§ 3.4). We then introduce two training paradigms: online PasoDoble (§ 3.5) and offline PasoDoble (§ 3.6).

### 3.1 OVERVIEW OF PASODOBLE

As illustrated in Figure 1, our PasoDoble framework comprises four components: the Proposer, the Solver, the Knowledge Base, and, for offline training, the Question Buffer. Both the Proposer and Solver are initialized from the same pretrained model and interact in an iterative loop. At each iteration, a knowledge piece is sampled from the Knowledge Base (①) to prompt the Proposer to generate question–answer (QA) pairs (②), which the Solver then attempts to solve with multiple solutions (③ – ④). The Solver receives a correctness reward based on agreement with the Proposer's answer (⑤). To assess question difficulty, we compute the Solver's accuracy per question (⑥) and define the Proposer's difficulty reward inversely with this accuracy (⑦), while a diversity reward encourages novel questions (⑧). These rewards are combined to yield the Proposer's final reward (⑨). Only valid questions with non-trivial difficulty are retained for Solver training (⑩). Both models are updated synchronously whenever at least one valid question is available (⑪), forming an online training loop. In offline training, the Proposer is first updated for several steps (⑪) while the Solver is frozen, and valid questions are stored in a Question Buffer (⑫). The Proposer is then frozen, and the Solver is updated on the buffered questions (⑬), constructing its training dataset.

## 3.2 THE KNOWLEDGE BASE

In PasoDoble, the Proposer is responsible for generating both questions and their ground-truth answers. This, however, would require the Proposer to be stronger than the Solver, to ensure that it provides correct answers even for problems challenging to the Solver. This assumption is unrealistic when both are initialized from the same base model. Consequently, we enrich the Proposer with high-quality external knowledge from a pretraining corpus $\mathcal{K}$. Specifically, we treat a segment of raw, unprocessed text from $\mathcal{K}$ as a unit of knowledge (e.g., text from a pretraining dataset). During training, the Proposer receives a sampled knowledge piece $k$ and is prompted to leverage it in generating QA pairs. This knowledge grounding enables the Proposer to produce challenging, diverse, and valid questions while ensuring the correctness of its answers without incurring the high costs of external supervision.

## 3.3 THE PROPOSER

The Proposer $\pi_p$ is an autoregressive language model tasked with generating QA pairs. At each iteration, it receives a knowledge piece $k$ sampled from the Knowledge Base $\mathcal{K}$ and generates a batch of $I$ question–answer pairs, denoted as $Q = \{(q_i, a_i^*)\}_{i=1}^I$. For each question $q_i$ with its ground-truth answer $a_i^*$, the Solver produces $J$ candidate solutions with final answers $\{a_{ij}\}_{j=1}^J$. The average passing rate of question $q_i$ is then computed as $p_i = \frac{1}{J} \sum_{j=1}^J \mathbb{I}(a_{ij} = a_i^*)$, where $\mathbb{I}(a_{ij} = a_i^*) = 1$ if $a_{ij}$ matches $a_i^*$, and $0$ otherwise. For RL training of the Proposer, PasoDoble's reward function incorporates a **difficulty** reward and a **diversity** reward.

**Difficulty reward.** The difficulty reward is inversely correlated with the Solver's passing rate: questions that the Solver finds harder (i.e., lower $p_i$) yield higher rewards. Formally, the difficulty reward is defined as $r_i^{\text{diff}} = 1.1 - p_i$. Note that even when $p_i = 1.0$, the Proposer still receives a positive reward of $0.1$, ensuring it is rewarded more than for an unsolvable question or one that does not follow the prompt instructions.

**Diversity reward.** Because diverse questions are essential for fostering the Solver's ability to develop generalizable reasoning skills, we introduce a diversity reward that penalizes similarity to recently generated questions. Formally, the diversity reward for question $q_i$ is defined as

$$r_i^{\text{div}} = 1 - \frac{\sum_{q_h \in H} \mathbb{I}(\text{Sim}(q_i, q_h))}{|H|},$$

where $H$ is a history buffer storing the most recent $|H|$ questions generated by the Proposer, and $\text{Sim}(q_i, q_h)$ indicates whether $q_i$ and $q_h$ are similar as measured vis the Jaccard index between the token sets, $\text{Tok}(q)$, of the two questions:

$$\text{Sim}(q_i, q_h) = \frac{|\text{Tok}(q_i) \cap \text{Tok}(q_h)|}{|\text{Tok}(q_i) \cup \text{Tok}(q_h)|}.$$

$\mathbb{I}(\text{Sim}(\cdot, \cdot))$ equals $1$ if the similarity between two questions exceeds the similarity threshold $\tau_{\text{sim}}$. Finally, the Proposer's reward is defined as a weighted combination of the question difficulty and diversity rewards:

$$r_i^p = \begin{cases} r_i^{\text{diff}} + w r_i^{\text{div}}, & p_i > \tau_{\text{low}} \text{ and } r_i^{div} \geq \tau_{\text{div}}, \\ 0, & \text{otherwise} \end{cases}$$

where $w$ is a weighting constant and $\tau_{\text{div}}$ is the diversity threshold. To prevent reward hacking by the Proposer (e.g., producing questions with incorrect answers), **we deem a question as *invalid* when its passing rate is too low** ($p_i \leq \tau_{\text{low}}$), and apply a hard clipping of the reward to 0. Additionally, we apply the same clipping if the diversity reward is too low ($r_i^{\text{div}} < \tau_{\text{div}}$), thereby further preventing the Proposer from generating overly repetitive questions. Thus, the Proposer is rewarded only when $p_i \geq \tau_{\text{low}}$ and $r_i^{\text{div}} \geq \tau_{\text{div}}$, to ensure the validity and diversity of the questions.

## 3.4 THE SOLVER

The Solver $\pi_s$ is an autoregressive language model responsible for solving the proposed questions. Its reward $r_{ij}^s$ is solely determined by **correctness**—that is, whether its predicted answer $a_{ij}$ matches the Proposer's ground-truth answer $a_i^*$: $r_{ij}^s = \mathbb{I}(a_{ij} = a_i^*)$.

### 3.5 ONLINE PASODOBLE TRAINING

In the online version of PasoDoble, at each iteration we update both the Proposer $\pi_p$ and the Solver $\pi_s$ whenever at least one valid question is available; otherwise, PasoDoble proceeds to the next iteration. During an update, the Proposer $\pi_p$ is trained on the set $Q = \{(q_i, a_i^*)\}_{i=1}^I$ with corresponding rewards $r^p = \{r_i^p\}_{i=1}^I$. The objective of the Proposer in the online setting is formally defined as

$$\mathcal{J}_{\mathrm{on}}(\pi_p) := \max_{\pi_p} \ \mathbb{E}_{k \in \mathcal{K}, \ (q_i, a_i^*) \sim \pi_p(\cdot | k)} \left[ r_i^p \right]. \tag{1}$$

For the Solver $\pi_s$, we retain only *valid* questions ($p_i \geq \tau_{\mathrm{low}}$) and further discard those with $p_i = 1$, since they provide no learning signal. We define a binary indicator $\mathrm{ret}_i$, which equals 1 if question $q_i$ is retained and 0 otherwise. Let $\mathbb{E}[\mathrm{ret}]$ denote the expectation of $\mathrm{ret}_i$ over knowledge and sample distributions. The Solver's objective is then:

$$\mathcal{J}_{\mathrm{on}}(\pi_s) := \max_{\pi_s} \ \mathbb{E}_{k \in \mathcal{K}, \ (q_i, a_i^*) \sim \pi_p(\cdot | k)} \left[ \frac{\mathrm{ret}_i}{\mathbb{E}[\mathrm{ret}]} \mathbb{E}_{a_{ij} \sim \pi_s(\cdot | q_i)} \left[ r_{ij}^s \right] \right], \tag{2}$$

$$= \min_{\pi_s} \ \mathbb{E}_{k \in \mathcal{K}, \ (q_i, a_i^*) \sim \pi_p(\cdot | k)} \left[ \frac{\mathrm{ret}_i}{\mathbb{E}[\mathrm{ret}]} \left( r_i^p \right)' \right]. \tag{3}$$

where $\left( r_i^p \right)' = r_i^{\mathrm{diff}}$ denotes the Proposer's difficulty reward. The normalization factor $\frac{\mathrm{ret}_i}{\mathbb{E}[\mathrm{ret}]}$ ensures that the retained questions are reweighted to form a proper distribution. This formulation highlights that the Solver's objective is (partially) zero-sum with respect to the Proposer's. Both objectives can be optimized using standard reinforcement learning algorithms such as PPO (Schulman et al., 2017) or GRPO (Shao et al., 2024a). The online training algorithm is presented in Appendix F.

### 3.6 OFFLINE PASODOBLE TRAINING

To improve the training stability, we also investigate an offline training paradigm. In each iteration, we first freeze the Solver and update the Proposer $\pi_p$ for $T_P$ steps using Eq. 1. During this process, we collect questions where $\tau_{\mathrm{low}} \leq p_i < 1$ and store the remaining ones in the question buffer $\mathcal{B}$. Next, we freeze the Proposer, and train the Solver for $T_S$ steps. In each step, we sequentially replay a set of questions $Q'$ in the question buffer to the Solver, and update the Solver $\pi_s$:

$$\mathcal{J}_{\mathrm{off}}(\pi_s) := \max_{\pi_s} \ \mathbb{E}_{(q_i, a_i^*) \sim Q', \ a_{ij} \sim \pi_s(\cdot | q_i)} \left[ r_{ij}^s \right]. \tag{4}$$

We loop over $\mathcal{B}$ in a circular fashion in case $\mathcal{B}$ is exhausted at any time point. Note that R-Zero can be regarded as a variant of our offline PasoDoble, which runs three iterations with a replay question buffer size of $|\mathcal{B}| = 8,000$. The offline training algorithm is presented in Appendix G.

## 4 EXPERIMENTAL SETUP

**Training Details.** We experiment with Qwen3-0.6B-Base, Qwen3-1.7B-Base, Qwen3-4B-Base, Qwen2.5-0.5B-Base, Qwen2.5-1.5B-Base, and Qwen2.5-3B-Base models (Qwen, 2025a;b) as both the Proposer and the Solver. To improve sampling efficiency and stabilize the early stages of training, we first conduct a cold-start supervised finetuning (SFT) stage. This step trains both the Proposer and the Solver to generate responses in the required format, for example, producing a reasoning trace followed by the problem and/or the final answer. Details of the SFT dataset are provided in Appendix A, and the formatting instructions are shown in Tables 11 and 12. After cold-start, we train both the Proposer and the Solver using GRPO (Shao et al., 2024b), and conduct experiments under both online and offline PasoDoble settings. In all experiments, we set the passing rate threshold to $\tau_{\mathrm{low}} = 0.2$, the diversity reward weight to $w = 0.2$, the diversity clipping threshold to $\tau_{\mathrm{div}} = 0.3$, the number of Proposer generations to $I = 6$, and the number of Solver generations to $J = 6$. Additional hyperparameters are listed in Appendix B.

**Knowledge Base.** We use the MegaMath-Pro-Max dataset (Wang et al., 2025) as the knowledge base $\mathcal{K}$, which contains millions of high-quality mathematical corpora. To accommodate computational resource constraints, we further filter out samples exceeding 1,024 tokens.

**Baselines.** We compare the trained Solver against two baselines: (1) the base model, and (2) the Solver after cold-start.

| Method | AIME 24 | AIME 25 | AMC | GSM8K | MATH | OlympiadBench | AVG |
|---|---|---|---|---|---|---|---|
| *Qwen2.5-0.5B* | | | | | | | |
| Base | 0.00 | **0.56** | 5.83 | 35.94 | **22.87** | **4.74** | 11.66 |
| Coldstart | 0.00 | 0.00 | 1.67 | 23.04 | 9.27 | 2.05 | 6.01 |
| Online PasoDoble | **0.56** | 0.00 | **7.92** | **40.26** | 19.63 | 4.20 | **12.10** |
| Offline PasoDoble | 0.00 | **0.56** | 3.75 | 34.52 | 16.63 | 3.73 | 9.87 |
| *Qwen3-0.6B* | | | | | | | |
| Base | 1.67 | 0.00 | 15.00 | 56.67 | **39.93** | **11.63** | 20.82 |
| Coldstart | 1.67 | 0.00 | 11.25 | 43.39 | 22.27 | 6.27 | 14.14 |
| Online PasoDoble | 1.11 | 1.11 | 19.17 | **64.03** | 38.70 | 11.53 | 22.61 |
| Offline PasoDoble | **2.78** | **1.67** | **19.58** | 63.96 | 39.00 | 11.53 | **23.09** |
| *Qwen2.5-1.5B* | | | | | | | |
| Base | 1.67 | 0.00 | 12.92 | 64.68 | 39.90 | 10.07 | 21.54 |
| Coldstart | 1.11 | 0.00 | 15.83 | 55.88 | 33.50 | 9.16 | 19.24 |
| Online PasoDoble | **3.89** | **2.22** | 20.83 | 72.13 | 45.17 | **16.27** | **26.75** |
| Offline PasoDoble | 0.56 | 1.11 | **23.75** | **72.83** | **46.27** | 13.80 | 26.38 |
| *Qwen3-1.7B* | | | | | | | |
| Base | 2.22 | 1.67 | 19.58 | 74.54 | 48.73 | 14.32 | 26.84 |
| Coldstart | 1.67 | 2.78 | 22.92 | 60.82 | 44.53 | 15.04 | 24.63 |
| Online PasoDoble | 6.11 | **7.78** | 37.92 | 83.13 | 66.67 | 28.00 | 38.27 |
| Offline PasoDoble | **7.22** | 7.22 | **40.83** | **84.98** | **68.50** | **28.79** | **39.59** |
| *Qwen2.5-3B* | | | | | | | |
| Base | 1.67 | 0.00 | 16.25 | 69.65 | 45.80 | 14.35 | 24.62 |
| Coldstart | 1.11 | 1.11 | 13.75 | 64.66 | 40.67 | 12.42 | 22.19 |
| Online PasoDoble | **5.56** | **3.33** | **30.42** | **82.26** | **58.70** | **22.59** | **33.81** |
| Offline PasoDoble | **5.56** | 2.78 | 26.67 | 81.48 | 55.67 | **22.59** | 32.46 |
| *Qwen3-4B* | | | | | | | |
| Base | 6.11 | 2.78 | 33.33 | 84.07 | 61.37 | 23.98 | 35.27 |
| Coldstart | 13.33 | 13.89 | 41.25 | 83.04 | 67.07 | 30.64 | 41.54 |
| Online PasoDoble | **18.89** | **18.89** | 53.33 | 91.82 | **82.17** | **42.27** | **51.23** |
| Offline PasoDoble | 17.78 | 16.67 | **55.42** | **91.91** | 79.63 | 40.81 | 50.37 |

Table 1: Main experiment results. **bold** = best, underlined = second best within each model group. Results on each dataset are the average pass@1 accuracy. AVG is the average across the benchmarks. Note that Base models are evaluated with 4-shot prompting, and other models use 0-shot prompting.

**Benchmarks.** We evaluate baselines and PasoDoble on AIME 2024, AIME 2025, AMC 23, GSM8K (Cobbe et al., 2021), MATH-500 (Lightman et al., 2023), and OlympiadBench (He et al., 2024). We sample six responses per question and report pass@1. Base models are evaluated with 4-shot in-context learning, while other models use the zero-shot prompt shown in Table 12. Additional details are provided in Table 5.

## 5 RESULTS AND ANALYSIS

In this section, we first present the performance of PoseDoble and the baselines on math benchmarks (§ 5.1). We then analyze the training dynamics of the Solver (§ 5.2), analyze decisions that govern the training of the Proposer (§ 5.3), and conduct ablation study to understand the role of the knowledge base, the importance of training the Proposer, and the need for ground-truth answers from the Proposer (§ 5.4).

### 5.1 MAIN RESULTS IN MATH REASONING

**PasoDoble improves model performance on math benchmarks, particularly for the Qwen2.5-1.5B/3B and Qwen3-1.7B/4B models. Furthermore, the gains from PasoDoble consistently increase with model scale.** The results of our experiments are presented in Table 1, Except for the smallest model (Qwen2.5-0.5B), PasoDoble consistently outperforms both the base and cold-start models, often substantially, with performance gains generally increasing with model size. Relative to the base models, average pass@1 accuracy changes by -2 points (0.5B), +2 points (0.6B), +5

points (1.5B), +11 points (1.7B), and +9 points (3B), and +16 points (4B) after online training, and by -2 points (0.5B), +2 points (0.6B), +5 points (1.5B), +13 points (1.7B), and +8 points (3B), and +15 points (4B) after offline training. Compared with cold-start models, online PasoDoble yields gains ranging from +6 points (0.5B) to +14 points (1.7B), while offline PasoDoble achieves improvements of +4 points (0.5B) to +15 points (1.7B). Since online and offline variants perform comparably, instability does not appear to be a major concern, though larger-scale experiments are needed to confirm this (see Appendix C). As in prior work (Huang et al., 2025b), PasoDoble still struggles with the most challenging datasets (AIME) on small models. Nevertheless, the notable improvement observed on Qwen3-4B (a 12-point gain) is encouraging, suggesting that future work should explore whether performance continues to scale with larger models, whether stronger knowledge bases can yield further gains, or whether fundamental limitations of the method remain.

## 5.2 TRAINING DYNAMICS OF THE SOLVER

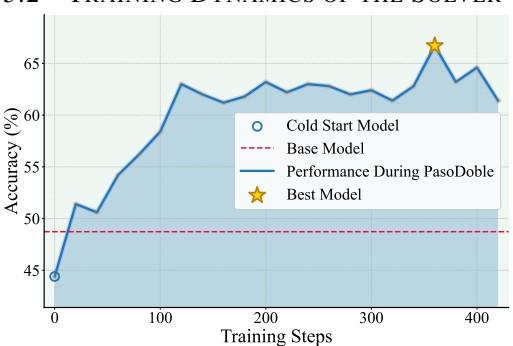
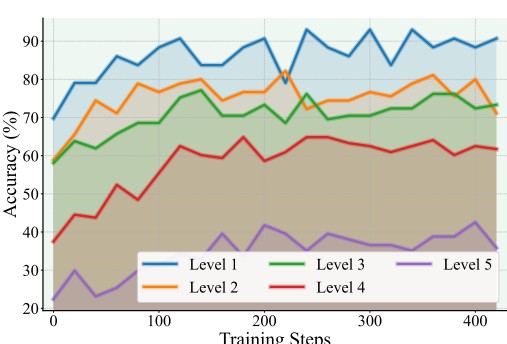

(a) Overall accuracy on MATH-500.

(b) Per-level accuracy on MATH-500.

Figure 2: Training Dynamics of the Solver

We analyze the Solver's performance across different training steps using Qwen3-1.7B under the online training paradigm, evaluated on MATH-500 (Figure 2). As shown in Figure 2a, the Solver's overall accuracy improves from 44.53% to 66.67%. It surpasses the base model before step 20 and exhibits a sharp increase between steps 0 and 120, reaching approximately 63% accuracy before plateauing. Thereafter, the Solver develops more slowly and fluctuates between 61% and 67%. The Solver achieves its best performance of 66.67% at step 320. Importantly, PasoDoble is able to sustain improvements during training in both the online and offline settings for hundreds of update steps, showing a much stronger scaling capacity than the related dual-play training of R-Zero (Huang et al., 2025b). We further analyze the Solver's progress across varying difficulty levels in MATH-500, aiming to identify which types of problems benefit most from training (Figure 2b). We find that Levels 1, 2, and 4 exhibit the most substantial improvements, each gaining about 20 percentage points in accuracy throughout training. Level 3 shows a slightly smaller improvement of around 16 points, while Level 5 achieves the most modest gain of approximately 12 points. Therefore, **our PasoDoble training substantially enhances the LLM's math reasoning performance for overall accuracy and for different problem difficulty levels**.

## 5.3 ANALYSIS OF THE PROPOSER

In this section, we analyze: (**1) Quality of Proposer's QA pairs**: how many questions generated by the Proposer have correct ground-truth answers, and how does this proportion vary when we incorporate the threshold $\tau_{\text{low}}$? (**2) Choice of $\tau_{\text{low}}$**: Is setting passing rate threshold $\tau_{\text{low}} = 0.2$ an optimal choice? The analysis is again based on Qwen3-1.7B under the online training paradigm, evaluated on MATH-500.

**Proposer's QA pair quality.** We sample 100 QA pairs from each Qwen3-1.7B offline checkpoint spanning steps 0 to 200. For each sampled QA pair, we use GPT-5 Mini to generate a reference answer and then compute the proportion of cases in which the Proposer's answer matches this reference. In Figure 3a, we observe that QA pairs with $p_i > \tau_{\text{low}}$ achieve much higher accuracy (90.17% for $p_i = 1$ and 72.98% for $\tau_{\text{low}} < p_i < 1$), whereas those with $p_i \leq \tau_{\text{low}}$ have considerably lower accuracy (30.65%), indicating that most contain incorrect ground-truth answers. For the Proposer, this implies that our decision to assign a reward of zero to the Proposer whenever $p_i \leq \tau_{\text{low}}$ effectively

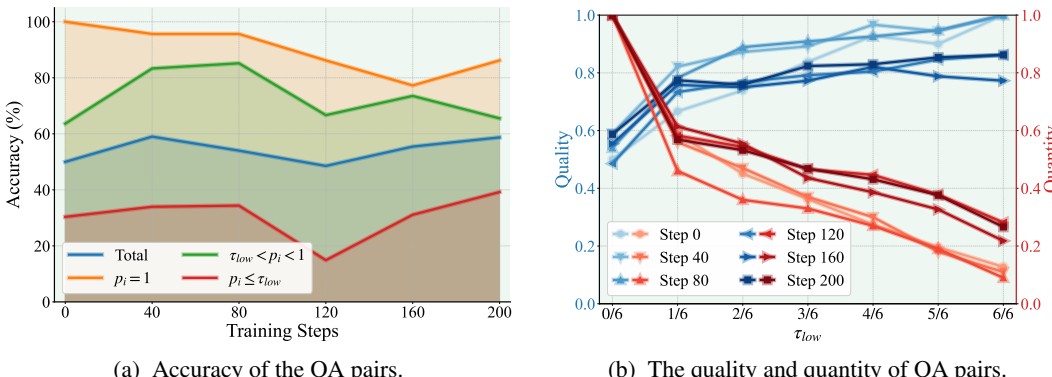

(a) Accuracy of the QA pairs.

(b) The quality and quantity of QA pairs.

Figure 3: Analysis of the Proposer.

discourages the generation of low-quality QA pairs. For the Solver, since we only retain questions where $\tau_{\text{low}} < p_i < 1$, which predominantly (73%) contain correct ground-truth answers, we effectively filter out low-quality QA pairs. Therefore, **the Proposer's QA pairs are of high quality after filtering, allowing them to correctly guide the training of the Proposer and Solver**.

Additionally, we observe a clear upward trend in overall accuracy as training progresses (rising from 50% at step 0 to 58.72% at step 200), suggesting that the Proposer steadily improves at generating higher-quality QA pairs.

**Choice of $\tau_{\text{low}}$.** An ideal $\tau_{\text{low}}$ should simultaneously maximize two factors: quantity (i.e., the number of QA pairs retained after filtering with $\tau_{\text{low}}$; if too many are filtered, sampling becomes inefficient) and quality (i.e., the proportion of retained pairs that have correct answers). Figure 3b illustrates how quantity and quality change as a function of $\tau_{\text{low}}$. We fix the number of attempts for both the Proposer and the Solver to 6 and retain questions for which $p_i \geq \tau_{\text{low}}$ (not $p_i > \tau_{\text{low}}$ as what we do in training). By varying $\tau_{\text{low}}$, we observe the trade-off between quantity and quality across Proposer checkpoints (steps 0–200). A threshold between $1/6$ and $2/6$ appears most effective, retaining roughly 50% of the QA pairs while achieving around 80% accuracy throughout training. Setting $\tau_{\text{low}}$ higher yields diminishing returns: for example, at $3/6$ the retention rate drops to about 80% of that at $2/6$, meaning $1.25\times$ more sampling is required to obtain the same number of correct QA pairs, while also increasing false negatives (discarding challenging but correct questions). Conversely, setting $\tau_{\text{low}}$ too low (e.g., $0/6$, no filtering) results in quality below 60%. These results suggest that **choosing $\tau_{\text{low}} = 0.2$ offers a reasonable balance between question quantity and quality**.

## 5.4 Ablation Study

We evaluate the effectiveness of individual aspects of PasoDoble, including the role of the knowledge base $\mathcal{K}$ (§ 5.4.1), training the Proposer (§ 5.4.2), and the need (or not) for ground-truth answers generated by the Proposer (§ 5.4.3).

### 5.4.1 Removing the Knowledge Base

To analyze the effect of external knowledge, we remove the knowledge base, forcing it to generate questions without external supervision. As shown in Table 2 (row labeled w/o $\mathcal{K}$), we observe that **for smaller models (Qwen2.5-0.5B and Qwen3-0.6B), removing knowledge surprisingly improves the Solver's performance, whereas for larger models (Qwen2.5-1.5B/3B and Qwen3-1.7B/4B), it has the opposite effect**. One possible explanation is that much of the external knowledge lies beyond the comprehension capacity of smaller models. When compelled to compose questions based on such material, the Proposer struggles to generate correct ground-truth answers, leading to lower question quality and, consequently, reduced Solver performance. In contrast, when composing without external supervision, the Proposer has the flexibility to generate questions it can reliably solve using its internal knowledge, thereby yielding cleaner supervision for the Solver.

| Method | AIME24 | AIME25 | AMC | GSM8K | MATH | OlympiadBench | AVG |
|---|---|---|---|---|---|---|---|
| *Qwen2.5-0.5B* | | | | | | | |
| Online PasoDoble | **0.56** | 0.00 | **7.92** | 40.26 | 19.63 | 4.20 | 12.10 |
| w/o $\mathcal{K}$ | **0.56** | 0.00 | 6.67 | **42.63** | **21.47** | **5.14** | **12.75** |
| w/ $\pi_p$ frozen | 0.00 | 0.00 | 3.75 | 33.07 | 14.43 | 3.41 | 9.11 |
| *Qwen3-0.6B* | | | | | | | |
| Online PasoDoble | 1.11 | **1.11** | **19.17** | 64.03 | 38.70 | 11.53 | 22.61 |
| w/o $\mathcal{K}$ | **1.67** | **1.11** | 17.92 | **64.32** | **44.50** | **13.16** | **23.78** |
| w/ $\pi_p$ frozen | 1.11 | 0.00 | 17.08 | 62.36 | 38.03 | 10.74 | 21.55 |
| *Qwen2.5-1.5B* | | | | | | | |
| Online PasoDoble | **3.89** | **2.22** | 20.83 | **72.13** | **45.17** | **16.27** | **26.75** |
| w/o $\mathcal{K}$ | 2.22 | **2.22** | **24.17** | 70.94 | 44.53 | 14.91 | 26.50 |
| w/ $\pi_p$ frozen | 1.11 | 0.56 | 18.33 | 68.54 | 42.57 | 12.96 | 24.01 |
| *Qwen3-1.7B* | | | | | | | |
| Online PasoDoble | 6.11 | **7.78** | **37.92** | **83.13** | **66.67** | **28.00** | **38.27** |
| w/o $\mathcal{K}$ | **9.44** | 6.67 | 34.17 | 81.54 | 63.73 | 26.67 | 37.04 |
| w/ $\pi_p$ frozen | 5.56 | 5.00 | 34.58 | 81.73 | 64.03 | 25.38 | 36.05 |
| *Qwen2.5-3B* | | | | | | | |
| Online PasoDoble | **5.56** | **3.33** | **30.42** | **82.26** | **58.70** | **22.59** | **33.81** |
| w/o $\mathcal{K}$ | 5.00 | 2.78 | 25.42 | 76.46 | 54.43 | 21.21 | 30.88 |
| w/ $\pi_p$ frozen | **5.56** | 2.22 | 29.58 | 80.06 | 56.50 | 22.47 | 32.73 |
| *Qwen3-4B* | | | | | | | |
| Online PasoDoble | **18.89** | **18.89** | **53.33** | **91.82** | **82.17** | **42.27** | **51.23** |
| w/o $\mathcal{K}$ | 17.22 | 13.33 | 48.33 | 86.92 | 75.40 | 35.85 | 46.18 |
| w/ $\pi_p$ frozen | 17.78 | 13.89 | 49.17 | 89.15 | 75.17 | 35.85 | 46.84 |

Table 2: Ablation study results. w/o $\mathcal{K}$: Solver trained with a Proposer that does not have external knowledge. w/ $\pi_p$ frozen: Solver trained with a frozen Proposer.

| Method | AIME24 | AIME25 | AMC | GSM8K | MATH | OlympiadBench | AVG |
|---|---|---|---|---|---|---|---|
| PasoDoble | **6.11** | **7.78** | **37.92** | **83.13** | **66.67** | **28.00** | **38.27** |
| w/ Full. Rand Rwd. | 0.00 | 0.00 | 0.42 | 2.88 | 1.07 | 0.10 | 0.75 |
| w/ Part. Rand Rwd. | 4.44 | 5.56 | 28.33 | 78.99 | 55.53 | 20.20 | 32.18 |

Table 3: Results of applying random rewards.

### 5.4.2 FROZEN PROPOSER

Another important question is whether it is necessary to train the Proposer. To examine this, we train the Solver with a frozen Proposer. The results, shown in Table 2 (row labeled w/ $\pi_p$ frozen), indicate **a consistent performance drop across models and benchmarks**. This outcome is expected: when the Proposer is frozen, the difficulty and diversity of its generated questions stagnates, causing the Solver to quickly saturate. As a result, the Solver's performance becomes upper-bounded by the behavior of the initial Proposer. These findings highlight the importance of allowing the Proposer to co-evolve with the Solver in order to achieve continued performance gains.

### 5.4.3 REMOVING THE PROPOSER'S ANSWERS

We investigate the impact of removing the Proposer's answers, such that only its *questions* are used. Since the answers are no longer visible to the Solver, we replace the Solver's correctness verifier with a random reward $r_{ij}^s \sim Bernoulli(0.5)$, which returns 1 or 0 uniformly at random (Fully Rand Rwd). This also induces a corresponding randomness in the Proposer's difficulty reward $r_i^{\text{diff}}$. Furthermore, we add an enhanced version where the Solver's reward is always 0 if its format is incorrect (i.e., no answer box), and random otherwise (Partial Rand Rwd).

Prior work (Shao et al., 2025) suggests that even random rewards may yield non-trivial improvements, which shows potential for these settings to work. We conduct the experiments on Qwen3-1.7B model using the online training paradigm. As shown in Table 3, training with fully random rewards drives the Solver's average accuracy on all math benchmarks to nearly zero. Even if we

| Method | AIME24 | AIME25 | AMC | GSM8K | MATH | OlympiadBench | AVG |
|--------|--------|--------|-----|-------|------|---------------|-----|
| PasoDoble | 6.11 | **7.78** | **37.92** | **83.13** | **66.67** | **28.00** | **38.27** |
| w/o $r^{\text{div}}$ | **8.33** | 2.22 | 30.83 | 82.08 | 60.47 | 24.07 | 34.67 |

Table 4: Results of removing diversity reward.

force the Solver to respond in the correct format (Part. Rand Rwd), its accuracy still decreases significantly. The sharp contrast with our original setting demonstrates that **the Solver benefits substantially from learning from Proposer's answers**. Moreover, as analyzed in § 5.3, these answers are predominantly correct after filtering.

### 5.4.4 REMOVING THE DIVERSITY REWARD

We study the effect of removing the diversity reward $r^{\text{div}}$ from the Proposer's training, in the Qwen3-1.7B online setting. As shown in Table 4, **eliminating diversity rewards leads to consistent performance degradation across math benchmarks**, with an average drop of approximately 4 points. We leave a more comprehensive exploration of diversity-oriented reward design, as well as a qualitative analysis of the questions generated with and without this term, to future work.

## 6 DISCUSSION, LIMITATIONS, AND CONCLUSION

PasoDoble demonstrates that a dual-play framework, in which a Proposer and Solver co-evolve through adversarial training, can be used to substantially improve LLMs' reasoning abilities. By rewarding the Proposer only for generating valid and challenging questions that push the Solver's limits, and rewarding the Solver for solving them correctly, our method fosters sustained competition and mutual evolution. Moreover, by enriching the Proposer with high-quality pre-training knowledge, PasoDoble serves as a bridge between pre-training and post-training, leveraging existing pre-training knowledge to enhance reasoning in the post-training stage. Compared to self-play, dual-play introduces complementary specialization between two distinct roles, opening a new perspective for LLM training. Extending this idea, one could imagine triple-play with three coordinated roles (e.g., proposer–solver–verifier) or even four-play with additional agents such as critics or knowledge retrievers, enabling richer forms of collaboration and competition among multiple LLMs.

At the same time, our study highlights important limitations. Performance gains for PasoDoble are less pronounced on small models; for example, Qwen3-0.6B-Base improves by only about 3%, while Qwen2.5-0.5B-Base even experiences a 1.5% drop, suggesting limited effectiveness at sub-1B scales. Furthermore, when applied to out-of-domain tasks such as GPQA, PasoDoble does not yield clear improvements, indicating its benefits may be domain-specific (See Appendix E for details). Finally, despite incorporating external knowledge, we observe that training still saturates after a few hundred update steps, pointing to the need for mechanisms that sustain long-term evolution.

In conclusion, our findings show both the potential and the challenges of dual-play frameworks. PasoDoble opens a promising direction for training LLMs, but scaling to larger models, extending to diverse domains, and mitigating saturation remain important avenues for future research.

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

## A  COLD-START

We perform a cold-start SFT for both the Proposer and the Solver to align them with the required output format as show in Table 11. To construct the Proposer's SFT dataset, we prompt Qwen3-14B to generate candidate QA pairs and retain only those with the correct format and the answer matches that of GPT5-mini. For the Solver's dataset, we feed the generated problems and keep only the responses whose answers also match GPT5-mini. This yields 2K examples for each role. Both models are then trained for 5 epochs with a learning rate of 1e-5 on their respective datasets. For conciseness, Proposer outputs are truncated at 6K tokens and Solver outputs at 5K tokens. Note that, we conduct separate SFT training on the Proposer for the $w/o \mathcal{K}$ settings, since the Proposer uses a different prompt without knowledge.

## B  HYPERPARAMETERS

We show the hyperparameters in Table 5.

|  | Hyperparameter | Value |
|---|---|---|
| Proposer | Max Prompt Length | 1280 |
|  | Max Completion Length | 6144 |
|  | Num Generations $I$ | 6 |
|  | Learning Rate | 1e-6 |
|  | KL coef $\beta$ | 0.0 |
|  | Sampling Temperature | 0.6 |
|  | Sampling Top p | 1.0 |
| Solver | Max Prompt Length | 512 |
|  | Max Completion Length | 6144 |
|  | Num Generations $J$ | 6 |
|  | Learning Rate | 1e-6 |
|  | KL coef $\beta$ | 0.0 |
|  | Sampling Temperature | 0.6 |
|  | Sampling Top p | 1.0 |
| Global | $\tau_{\text{low}}$ | 0.2 |
|  | $\tau_{\text{sim}}$ | 0.3 |
|  | $\tau_{\text{div}}$ | 0.3 |
|  | $w_{\text{div}}$ | 0.2 |
|  | Size of history buffer $H$ | 100 |
| Online | Max Training Steps | 600 |
| Offline | Max Num Iterations | 60 |
|  | Per Iteration Proposer Steps $T_P$ | 10 |
|  | Per Iteration Solver Steps $T_S$ | 5 |
| Evaluation | Max Context Length | 8192 |
|  | Sampling Temperature | 0.6 |
|  | Sampling Top p | 0.95 |
|  | Num Generations | 6 |

Table 5: List of hyperparameters.

## C  ANALYSIS OF ONLINE AND OFFLINE TRAINING STABILITY

In this section, we analyze the training stability of the online and offline paradigms, focusing on Qwen3-1.7B-Base.

Figure 4 compares the Proposers. After the initial steps, the per-batch reward means converge to similar levels in both settings, and the reward standard deviations are also comparable, with the exception of a spike around step 80 in the online setting.

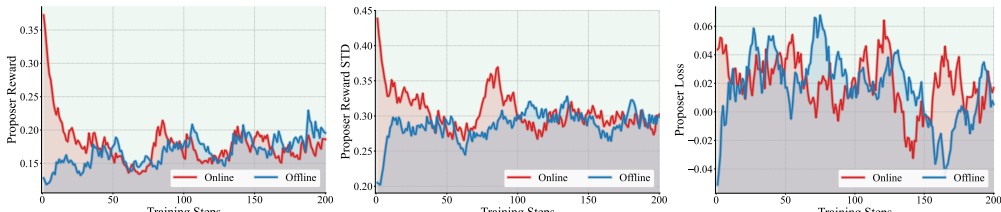

Figure 4: Comparison between online and offline training of the Proposer. Left: mean reward over the $I$ questions generated from each knowledge piece $k$; Middle: standard deviation of the Proposer's reward per $k$; Right: Proposer training loss. All curves are smoothed using an exponential moving average with a factor of 0.9. [1]

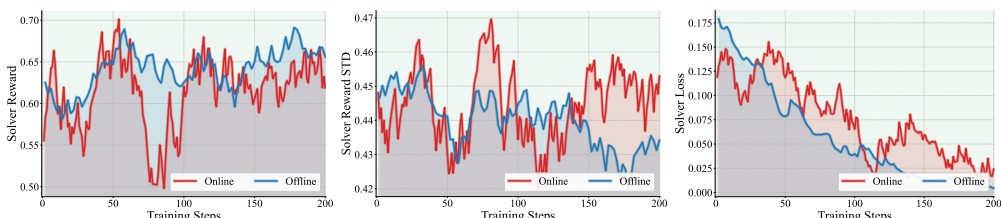

Figure 5: Comparison of online and offline training for the Solver. Left: mean reward over the $J$ responses to each question $q_i$; Middle: standard deviation of rewards per $q_i$; Right: Solver training loss. All curves use an exponential moving average with smoothing factor 0.9. For the offline setting, the $x$-axis ("Training Steps") is rescaled so that a value of $x$ indicates the Proposers paired with both the online and offline Solvers have each been trained for $x$ steps.

In contrast, Figure 5 highlights more pronounced differences in the Solvers. Under online training, the Solver's per-question reward mean and standard deviation fluctuate substantially, whereas in offline training the reward exhibits a steady upward trend and the standard deviation decreases consistently. In addition, the training loss declines more smoothly in the offline setting.

|  |  | Reward Mean | Reward STD | Loss |
|---|---|---|---|---|
| Proposer | Online | 0.0814 | **0.0845** | **0.0825** |
|  | Offline | **0.0830** | 0.0782 | 0.0766 |
| Solver | Online | **0.1745** | **0.0460** | **0.0847** |
|  | Offline | 0.1105 | 0.0338 | 0.0536 |

Table 6: Standard deviation of the metrics across training steps.

Table 6 reports the standard deviation of each metric across training steps. Overall, online training exhibits higher variability for both the Proposer and the Solver, suggesting less stability. A more detailed investigation of these behavioral differences is left for future work.

## D   EVICTION OF QUESTIONS IN THE QUESTION BUFFER

During offline training, if too few new questions are added to the question buffer in each Proposer iteration, the Solver will be repeatedly trained on the same set of questions and may quickly overfit on them. To avoid overfitting, we evict the question in the buffer when either of the following

---

[1] The Proposer's initial reward mean and standard deviation differ markedly between online and offline training, which we suspect is due to randomness in the knowledge samples drawn from the knowledge base.

conditions is true: (1) the Solver achieves 100% accuracy on the question; (2) the Solver's passing rate on the question has not increased from its peak passing rate in the last $c = 3$ attempts. If $|\mathcal{B}|$ turns into 0, we stop the current Solver iteration early.

| Method | AIME24 | AIME25 | AMC | GSM8K | MATH | OlympiadBench | AVG |
|---|---|---|---|---|---|---|---|
| *Qwen3-1.7B* | | | | | | | |
| Base | 2.22 | 1.67 | 19.58 | 74.54 | 48.73 | 14.32 | 26.84 |
| Coldstart | 1.67 | 2.78 | 22.92 | 60.82 | 44.53 | 15.04 | 24.63 |
| Offline PasoDoble | **7.22** | **7.22** | **40.83** | **84.98** | **68.50** | **28.79** | **39.59** |
| w/ eviction | 5.56 | 4.44 | 27.08 | 79.16 | 58.07 | 22.10 | 32.74 |

Table 7: Results after adding question eviction mechanism.

As shown in Table 7, our method becomes less effective after introducing the eviction mechanism. We suspect that the hyperparameter $c$ in condition (2) may be too small, causing questions to be discarded prematurely before the Solver has a chance to make progress. A more thorough investigation of this issue is left for future work.

# E  OUT-OF-DOMAIN EVALUATIONS

We further assess PasoDoble against the baselines described in §4 on out-of-domain benchmarks, specifically GPQA (Rein et al., 2024) and SuperGPQA (Du et al., 2025). We run each benchmark five times and report the average pass@1. Base models are evaluated with 5-shot in-context learning, whereas all other models adopt the zero-shot prompt from Table 12. The results are presented in Table 8.

We find that PasoDoble yields only slight improvements for Qwen2.5-0.5B in both the online and offline settings, while providing gains of about 2.5 points for Qwen2.5-1.5B and Qwen3-4B under both settings compared to their base models. However, it slightly decreases performance for the remaining models. One possible explanation is that training with PasoDoble relies on high-quality pre-training math knowledge, which shifts the model distribution further toward the math domain. We leave a deeper investigation of this phenomenon to future work. In conclusion, PasoDoble does not improve performance on out-of-domain tasks.

# F  ONLINE PASODOBLE ALGORITHM

The online algorithm is shown in Algorithm 1.

# G  OFFLINE PASODOBLE ALGORITHM

The offline algorithm is shown in Algorithm 2.

| Method | GPQA | SuperGPQA | AVG |
|---|---|---|---|
| *Qwen2.5-0.5B* | | | |
| Base | 24.75 | 11.30 | 18.03 |
| Coldstart | 25.02 | 11.15 | 18.09 |
| Online PasoDoble | 25.31 | 12.82 | 19.07 |
| Offline PasoDoble | **26.12** | **13.37** | **19.75** |
| *Qwen3-0.6B* | | | |
| Base | **26.77** | **15.03** | **20.90** |
| Coldstart | 25.65 | 14.56 | 20.11 |
| Online PasoDoble | 25.39 | 13.07 | 19.23 |
| Offline PasoDoble | 26.63 | 14.04 | 20.34 |
| *Qwen2.5-1.5B* | | | |
| Base | 24.24 | 17.64 | 20.94 |
| Coldstart | 25.63 | 17.93 | 21.78 |
| Online PasoDoble | **28.52** | 18.84 | **23.68** |
| Offline PasoDoble | 27.62 | **19.10** | 23.36 |
| *Qwen3-1.7B* | | | |
| Base | 28.28 | **20.92** | **24.60** |
| Coldstart | 28.32 | 19.21 | 23.77 |
| Online PasoDoble | 28.78 | 20.16 | 24.47 |
| Offline PasoDoble | **29.31** | 19.43 | 24.37 |
| *Qwen2.5-3B* | | | |
| Base | 26.26 | **20.31** | **23.29** |
| Coldstart | 28.04 | 16.56 | 22.30 |
| Online PasoDoble | **29.06** | 16.65 | 22.86 |
| Offline PasoDoble | 27.50 | 17.51 | 22.51 |
| *Qwen3-4B* | | | |
| Base | 36.36 | 28.43 | 32.40 |
| Coldstart | 34.82 | 24.33 | 29.58 |
| Online PasoDoble | **39.20** | 30.67 | **34.94** |
| Offline PasoDoble | 38.34 | **30.82** | 34.58 |

Table 8: Out-of-domain results on GPQA and SuperGPQA. **Bold** denotes the best and underline the second-best within each model group. All values are pass@1 and rounded to two decimals; AVG is the arithmetic mean of GPQA and SuperGPQA.

---

**Algorithm 1** Online PasoDoble Training Paradigm

---

**Require:** Knowledge base $\mathcal{K}$; Proposer $\pi_p$; Solver $\pi_s$; thresholds $\tau_{\text{low}}, \tau_{\text{div}}$; History buffer $H$; weight $w$; number of the generations of the Proposer and the Solver $I, J$; iterations $T$

1: $H \leftarrow \emptyset$
2: **for** $t = 1$ to $T$ **do**
3:     $k \sim \text{Sample}(\mathcal{K})$                              $\triangleright$ ①: sample knowledge
4:     $Q \leftarrow \emptyset, \mathcal{D}_p \leftarrow \emptyset, \mathcal{D}_s \leftarrow \emptyset$
5:     **for** $i = 1$ to $I$ **do**                       $\triangleright$ ②: generate $I$ QA pairs
6:         $(q_i, a_i^*) \sim \pi_p(\cdot \mid k); Q \leftarrow Q \cup \{(q_i, a_i^*)\}$
7:         **for** $j = 1$ to $J$ **do**
8:             $a_{ij} \sim \pi_s(\cdot \mid q_i)$               $\triangleright$ ③ - ④: solver attempts
9:             $r_{ij}^s \leftarrow \mathbb{I}(a_{ij} = a_i^*)$        $\triangleright$ ⑤: compute solver reward
10:         **end for**
11:         $p_i \leftarrow \frac{1}{J} \sum_{j=1}^{J} r_{ij}^s$            $\triangleright$ ⑥: average the accuracy
12:         $r_i^{\text{diff}} \leftarrow \textit{DifficultyReward}(p_i, \tau_{low})$     $\triangleright$ ⑦: compute difficulty reward
13:         $r_i^{\text{div}} \leftarrow \textit{DiversityReward}(q_i, H)$      $\triangleright$ ⑧: compute diversity reward
14:         $r_i^p \leftarrow \textit{FinalReward}(r_i^{\text{diff}}, r_i^{\text{div}}, w, \tau_{div})$   $\triangleright$ ⑨: compute final proposer reward
15:         $\mathcal{D}_p \leftarrow \mathcal{D}_p \cup \{(q_i, a_i^*, r_i^p)\}$
16:         **if** $\textit{Valid}(p_i, \tau_{low})$ *and* $p_i < 1$ **then**      $\triangleright$ ⑩: verify the question
17:             $\mathcal{D}_s \leftarrow \mathcal{D}_s \cup \{(q_i, a_{ij}, r_{ij}^s)\}_{j=1}^{J}$
18:         **end if**
19:         $H \leftarrow H \cup \{q_i\}$; truncate $H$ to the most recent $|H|$ questions
20:     **end for**
21:     **if** $\mathcal{D}_s \neq \emptyset$ **then**                $\triangleright$ at least one valid question
22:         **Update Proposer** $\pi_p$ on $\mathcal{D}_p$ with rewards $r^p$ via RL    $\triangleright$ ⑪: optimize Eq. 1
23:         **Update Solver** $\pi_s$ on $\mathcal{D}_s$ with rewards $r^s$ via RL     $\triangleright$ ⑪: optimize Eq. 2
24:     **else**
25:         **continue**              $\triangleright$ skip updates if no valid question this iteration
26:     **end if**
27: **end for**

---

---

**Algorithm 2** Offline PasoDoble Training Paradigm

---

**Require:** Knowledge base $\mathcal{K}$; Proposer $\pi_p$; Solver $\pi_s$; thresholds $\tau_{\text{low}}, \tau_{\text{div}}$; history buffer $H$; weight $w$; Proposer/Solver generations $I, J$; iterations $T$; Proposer steps $T_P$; Solver steps $T_S$; question buffer $\mathcal{B}$

1:  $H \leftarrow \emptyset, \mathcal{B} \leftarrow \emptyset$
2:  **for** $t = 1$ to $T$ **do**
3:      **Freeze** $\pi_s$; **Unfreeze** $\pi_p$                         ▷ phase A: proposer update
4:      **for** $u = 1$ to $T_p$ **do**
5:          $k \sim \text{Sample}(\mathcal{K})$                         ▷ ① : sample knowledge
6:          $Q \leftarrow \emptyset, \mathcal{D}_p \leftarrow \emptyset, \mathcal{D}_s^{\text{temp}} \leftarrow \emptyset$
7:          **for** $i = 1$ to $I$ **do**                         ▷ ② : generate $I$ QA pairs
8:              $(q_i, a_i^*) \sim \pi_p(\cdot \mid k); Q \leftarrow Q \cup \{(q_i, a_i^*)\}$
9:              **for** $j = 1$ to $J$ **do**
10:                 $a_{ij} \sim \pi_s(\cdot \mid q_i)$                         ▷ ③ - ④ : solver attempts (frozen)
11:                 $r_{ij}^s \leftarrow \mathbb{I}(a_{ij} = a_i^*)$                         ▷ ⑤ : compute solver reward
12:             **end for**
13:             $p_i \leftarrow \frac{1}{J} \sum_{j=1}^{J} r_{ij}^s$                         ▷ ⑥ : average the accuracy
14:             $r_i^{\text{diff}} \leftarrow \textit{DifficultyReward}(p_i, \tau_{\text{low}})$                         ▷ ⑦ : compute difficulty reward
15:             $r_i^{\text{div}} \leftarrow \textit{DiversityReward}(q_i, H)$                         ▷ ⑧ : compute diversity reward
16:             $r_i^p \leftarrow \textit{FinalReward}(r_i^{\text{diff}}, r_i^{\text{div}}, w, \tau_{\text{low}}, \tau_{\text{div}})$         ▷ ⑨ : compute final proposer reward
17:             $\mathcal{D}_p \leftarrow \mathcal{D}_p \cup \{(q_i, a_i^*, r_i^p)\}$
18:             **if** $\textit{Valid}(p_i, \tau_{\text{low}})$ and $p_i < 1$ **then**                         ▷ ⑩ : verify the question
19:                 $\mathcal{B} \leftarrow \mathcal{B} \cup \{(q_i, a_i^*, used = 0, prev\_acc = p_i)\}$         ▷ ⑫ : store valid questions to buffer
20:                 $\mathcal{D}_s^{\text{temp}} \leftarrow \mathcal{D}_s^{\text{temp}} \cup \{(q_i, a_i^*, r_i^p)\}$
21:             **end if**
22:             $H \leftarrow H \cup \{q_i\}$; truncate $H$ to the most recent $|H|$ questions
23:         **end for**
24:         **if** $\mathcal{D}_s^{\text{temp}} \neq \emptyset$ **then**
25:             **Update Proposer** $\pi_p$ on $\mathcal{D}_p$ with rewards $r^p$ via RL                         ▷ ⑪ : optimize Eq. 1
26:         **else**
27:             continue
28:         **end if**
29:     **end for**
30:     **Freeze** $\pi_p$; **Unfreeze** $\pi_s$                         ▷ phase B: solver update
31:     **for** $v = 1$ to $T_S$ **do**
32:         $Q' \leftarrow \textit{Replay}(\mathcal{B})$                         ▷ ⑬ : sequential replay from buffer
33:         $\mathcal{D}_s \leftarrow \emptyset$
34:         **for all** $(q_i, a_i^*) \in Q'$ **do**
35:             **for** $j = 1$ to $J$ **do**
36:                 $a_{ij} \sim \pi_s(\cdot \mid q_i)$                         ▷ ③ - ④ : solver attempts
37:                 $r_{ij}^s \leftarrow \mathbb{I}(a_{ij} = a_i^*)$                         ▷ ⑤ : compute solver reward
38:             **end for**
39:             **if** Evict $q_i$ **then**
40:                 $\mathcal{B} \leftarrow \mathcal{B} \setminus \{(q_i, a_i^*, \cdot)\}$
41:             **end if**
42:             $\mathcal{D}_s \leftarrow \mathcal{D}_s \cup \{(q_i, a_{ij}, r_{ij}^s)\}_{j=1}^{J}$
43:         **end for**
44:         **Update Solver** $\pi_s$ on $\mathcal{D}_s$ with rewards $r^s$ via RL                         ▷ ⑪ : optimize Eq. 4
45:     **end for**
46:  **end for**

---

# H   ANALYSIS OF THE PROPOSER

In this section, we analyze: **(1) The type of external knowledge provided to the Proposer** (§ H.1); **(2) The association between the Proposer's generated questions and both external and internal knowledge** (§ H.2); **(3) The association between the Proposer's generated answers and both external and internal knowledge** (§ H.3); **(4) The dynamics of the Proposer's question difficulty, diversity, and sampling efficiency.** (§ H.4); and **(5) The effectiveness of the Proposer in solving math problems** (§ H.5). Unless otherwise noted, all analyses in this section are conducted under the offline training setting with Qwen3-1.7B. We choose the offline setting due to its greater training stability: compared to the online setting, it exhibits more consistent reward signals and smoother behavior across training checkpoints. This stability enables clearer observation of how Proposer dynamics evolve over time and facilitates more reliable comparative analysis. (See Appendix C for a detailed comparison between online and offline settings.)

## H.1   TYPES OF EXTERNAL KNOWLEDGE

We analyze the external knowledge used by the Proposer and employ GPT-5-mini to assign a type to each knowledge item (see Figure 14 for the prompt). The knowledge is grouped into five content-based categories, and individual items may be assigned to multiple categories.

**Concept/Theorem**. The knowledge describes a mathematical concept (e.g., the definition of a topic or notion) or a theorem (e.g., a statement about the properties of a mathematical object). Because many knowledge pieces are extracted excerpts, their descriptions may be incomplete, for example, presented only in natural language without formal notation or missing key details. We therefore further divide this category into complete and incomplete subtypes. A knowledge piece that describes one concept/theorem completely but another incompletely is counted in both subcategories.

| Type | Percentage |
|------|-----------|
| Concept/Theorem | 61.67 |
| Complete | 36.50 |
| Incomplete | 25.17 |
| Problem | 33.50 |
| Method | 52.50 |
| Other | 10.50 |

Table 9:  Knowledge type distribution.

**Problem**. The knowledge explicitly describes a mathematical problem to be solved. In many cases, such knowledge pieces are excerpts from homework assignments or problem sets. Some also originate from textbooks or Wikipedia entries describing conjectures or open problems.

**Method**. The knowledge describes a procedure or technique for solving a particular type or instance of a mathematical problem. In many cases, it also explicitly states the corresponding problem itself.

**Other**. Knowledge pieces that do not fall under any of the above categories are classified as Other.

We aggregate the knowledge pieces from checkpoints 0 to 200 and report the results in Table 9. Approximately 60% of the knowledge pieces describe mathematical concepts or theorems, of which roughly three-fifths are complete. In addition, 30% of the knowledge pieces pertain to problems and 50% pertain to solution methods. Only a small number fall into the "Other" category. As shown in our manual inspection in Appendix I, the QA pairs in this category primarily concern educational context (e.g., descriptions of math education programs, teaching experiences, or courses) rather than substantive mathematical content.

| Domain | Percentage |
|--------|-----------|
| Algebra | 17.83 |
| Number Theory | 7.83 |
| Geometry | 8.33 |
| Trigonometry | 1.83 |
| Calculus | 7.33 |
| Combinatorics | 4.00 |
| Probability/Statistics | 18.67 |
| Optimization | 0.33 |
| Other | 33.83 |

Table 10:  Knowledge domain distribution.

We further analyze which math areas the knowledge pieces fall into. In particular, we define the following math domains: algebra, number theory, geometry, trigonometry, calculus, combinatorics, probability/statistics, optimization, and other. We then prompt GPT-5-mini (See prompt in Figure 15) to categorize the knowledge pieces into one of the domains.

The results are shown in Table 10. The most common domains are Algebra, Probability/Statistics, and Other. Upon manual inspection, many of the knowledge pieces categorized under "Other" pertain to subjects such as chemistry, physics, or programming.

## H.2 KNOWLEDGE-QUESTION ASSOCIATION

To analyze how external knowledge influences the Proposer to generate questions, we prompt GPT-5 to examine both the Proposer's question and the provided external knowledge piece (See prompt in Figure 16). Our manual examination (Appendix I) shows that all the Proposer's questions are topically aligned with the provided external knowledge. We further categorize each question based on the source of its content:

**External.** The question and its conditions are directly extracted (i.e., copied) from the external knowledge, assuming the external knowledge already contains a question and provides all necessary conditions.

**Internal.** Neither the question nor its conditions are explicitly stated or implicitly derivable from the external knowledge. In this case, the external knowledge serves at most as a topical cue, and the question is generated entirely from the model's internal parametric knowledge.

**Both.** The question draws on both external and internal knowledge. That is, part of the question (or its conditions) is explicitly stated or implicitly derivable from the external knowledge, while other parts are not. This category includes cases where the question is a modified rephrasing of a problem described in the external knowledge (e.g., with certain conditions added, removed, or altered), or an instantiation of a general problem form in the knowledge

| Checkpoint | External | Internal | Both |
|---|---|---|---|
| 0 | 46.00 | 5.00 | 49.00 |
| 40 | 43.00 | 13.00 | 44.00 |
| 80 | 43.00 | 5.00 | 52.00 |
| 120 | 40.00 | 7.00 | 53.00 |
| 160 | 48.00 | 7.00 | 45.00 |
| 200 | 49.00 | 4.00 | 47.00 |

Table 11: Question source distributions across the checkpoints.

that requires the model to supply specific details. It also includes cases where solving the question depends on a concept, theorem, or method mentioned in the external knowledge, but additional internal knowledge is still needed. In such cases, the model must interpret the external knowledge and integrate it with its internal parametric knowledge to construct a coherent question and its conditions.

As shown in Table 11, across step 40 to 200, approximately 40–50% of the questions are directly copied from the external knowledge, while only 4–10% are generated purely from the Proposer's internal knowledge. The remaining 40–50% draw on both sources. This indicates that the Proposer is neither merely extracting questions from the knowledge base nor using the knowledge only as a loose topical cue. Rather, the two sources interact in a meaningful way during question generation.

We further examine whether the questions generated by the Proposer are recalled from pre-training or newly composed through reasoning during PasoDoble. Following the experiment setup of Wu et al. (2025), we prompt Qwen3-1.7B with the first $x\%$ of each question generated by the Proposer ($x \in 40, 60, 80$) and test whether it can reconstruct the remainder using greedy decoding without applying the chat template. For each checkpoint, we evaluate the same 100 questions and report Exact Match (EM) and ROUGE-L (Lin & Och, 2004) in Table 12. Let $q_{old}$ denote the original question and $q_{new}$ the generated continuation. ROUGE-L measures the proportion of the longest common subsequence between $q_{old}$ and $q_{new}$ relative to the length of $q_{new}$, capturing how much of $q_{new}$ overlaps with $q_{old}$. EM is 1 if $q_{new}$ exactly matches $q_{old}$, and 0 otherwise.

Compared to the results reported in Wu et al. (2025), the questions generated by our Proposer, both with external knowledge and without external knowledge, consistently exhibit lower EM and ROUGE-L scores. This indicates that the Proposer is not simply copying or slightly modifying existing questions but is composing new questions through genuine reasoning during PasoDoble training.

We further focus on the setting with external knowledge and use the setting without external knowledge as a baseline. Even when we provide the first 80% of the question tokens to the Proposer, it is able to regenerate the original question $q_{old}$ for only about 10% of the cases. This rate is sig-

| Method | Checkpoint | 80%-Problem | | 60%-Problem | | 40%-Problem | |
|---|---|---|---|---|---|---|---|
| | | ROUGE-L | EM | ROUGE-L | EM | ROUGE-L | EM |
| w/ $\mathcal{K}$ | 0 | 51.17 | 15.00 | 41.17 | 1.00 | 32.89 | 0.00 |
| | 40 | 49.30 | 12.00 | 39.15 | 0.00 | 32.77 | 0.00 |
| | 80 | 44.40 | 7.00 | 40.19 | 0.00 | 34.75 | 0.00 |
| | 120 | 50.30 | 10.00 | 40.83 | 2.00 | 34.92 | 0.00 |
| | 160 | 43.63 | 11.00 | 36.39 | 0.00 | 31.30 | 0.00 |
| | 200 | 50.55 | 12.00 | 36.45 | 1.00 | 33.07 | 0.00 |
| w/o $\mathcal{K}$ | 0 | 58.70 | 22.00 | 53.79 | 4.00 | 40.75 | 0.00 |
| | 40 | 62.88 | 22.00 | 53.20 | 2.00 | 45.72 | 0.00 |
| | 80 | 67.65 | 27.00 | 59.67 | 6.00 | 44.58 | 0.00 |
| | 120 | 66.19 | 27.00 | 57.10 | 7.00 | 45.92 | 1.00 |
| | 160 | 71.14 | 37.00 | 61.37 | 9.00 | 48.61 | 3.00 |
| | 200 | 67.17 | 31.00 | 55.95 | 6.00 | 45.48 | 2.00 |

Table 12: Exact Match (EM) and ROUGE-L scores of Qwen3-1.7B on questions generated by the Proposer at different training checkpoints, under varying prompt prefix ratios. All evaluations use greedy decoding and without the chat template.

nificantly lower than the rate observed in the setting without external knowledge. The observation suggests that most of the questions are not fully recoverable from the Proposer's internal parametric knowledge alone, and the Proposer relies on the external knowledge to generate them. This finding is consistent with the distribution presented in Table 11.

### H.3 KNOWLEDGE-ANSWER ASSOCIATION

We further examine how external knowledge influences the Proposer's answer (See prompt in Figure 17). Specifically, we evaluate whether the answer to a question can be derived solely from the external knowledge provided, or whether it additionally relies on its internal knowledge. Naturally, if both the question and answer are copied directly from the knowledge, the answer is considered derivable. To make this criterion slightly more permissive, we also treat an answer as derivable if it requires only minimal common-sense reasoning (e.g., basic arithmetic) beyond the external knowledge.

As shown in Table 13, except for checkpoint 40, more than half of all answers are derivable from external knowledge alone. This implies that, aside from the 20-30% of the questions that are copied from external knowledge (those under "external" in Table 11), there are an additional 20-30% of questions under the "both" category in Table 11 that are mere extensions of existing questions/concepts mentioned in the knowledge.

| Checkpoint | Derivable | Not Derivable |
|---|---|---|
| 0 | 65.00 | 35.00 |
| 40 | 44.00 | 56.00 |
| 80 | 57.00 | 43.00 |
| 120 | 51.00 | 49.00 |
| 160 | 54.00 | 46.00 |
| 200 | 60.00 | 40.00 |

Table 13: Derivability of the answers across the checkpoints.

### H.4 DYNAMICS OF QUESTION DIFFICULTY, DIVERSITY, AND SAMPLING EFFICIENCY

We analyze the dynamics of the Proposer's question difficulty, diversity, and sampling efficiency using checkpoints spanning steps 0 to 200, evaluated both with and without external knowledge. Again, we use the 100 QA pairs from each Proposer checkpoint to do the analysis.

**Dynamics of the question difficulty.** We report the average question difficulty and sampling efficiency in Figure 6a (with external knowledge) and Figure 6b (without external knowledge). We additionally report the distribution of difficulty levels in Figure 7a (with external knowledge) and

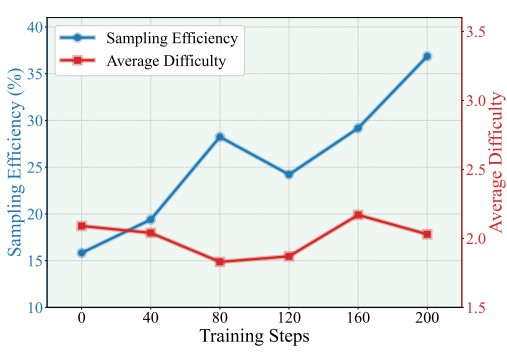 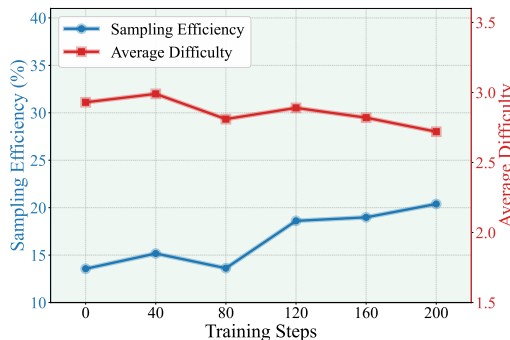

(a) Average question difficulty and sampling efficiency w/ $\mathcal{K}$.

(b) Average question difficulty and sampling efficiency w/o $\mathcal{K}$.

Figure 6: Analysis on average question difficulty and sampling efficiency.

Figure 7b (without external knowledge). The difficulty labels are assigned by GPT-5-mini using the prompt provided in Figure 13. For the setting with the external knowledge, as shown in Figure 6a, the average question difficulty fluctuates within a relatively narrow range (approximately 1.8–2.2) throughout training, indicating that the Proposer maintains a stable difficulty level when different external knowledge is provided. For the setting without external knowledge, as shown in Figure 6b, the average question difficulty begins at a relatively high level (around 2.9) and gradually decreases to approximately 2.7 as training progresses. This indicates that the Proposer initially generates more difficult questions but gradually shifts toward more moderate ones. A likely explanation is that difficult questions receive a reward of zero when the Solver's passing rate falls below the threshold $\tau_{\text{low}}$, discouraging the Proposer from continuing to generate such questions. Developing mechanisms to preserve or reintroduce hard but valid QA pairs during training, while at the same time avoiding reward hacking, remains an interesting direction for future work.

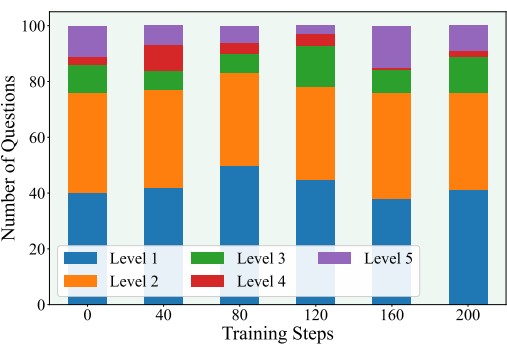 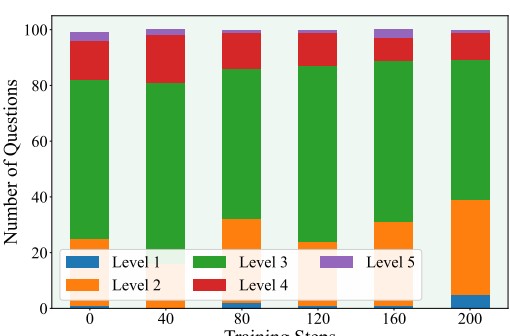

(a) Number of questions per-level w/ $\mathcal{K}$.

(b) Number of questions per-level w/o $\mathcal{K}$.

Figure 7: Analysis on number of questions per-level.

Moreover, the per-level distributions in Figure 7a (with external knowledge) and Figure 7b (without external knowledge) exhibit notable differences. When external knowledge is provided (Figure 7a), the Proposer generates a higher proportion of Level 1 and Level 5 questions. A plausible explanation is that some Level 5 questions arise when the Proposer directly extracts or adapts complex problem statements contained in the external knowledge, or when the base model already possesses sufficient internal knowledge to leverage these concepts, allowing it to formulate or recognize inherently difficult problems. In contrast, the increased prevalence of Level 1 questions may stem from the fact that the base model of the Proposer and the Solver do not fully internalize such external knowledge during pre-training; as a result, even relatively simple questions derived from the external knowledge may pose challenges for the Solver, leading the Proposer to favor simpler formulations that still yield positive rewards. When external knowledge is not provided (Figure 7b), the Proposer generates a substantially larger number of Level 3 questions, especially in the earlier checkpoints. Over time, the proportion of Level 3 questions decreases slightly, while Level 2 questions increase,

mirroring the downward trend in average difficulty discussed above. Level 1 and Level 5 questions remain consistently scarce throughout training. This pattern suggests that, in the absence of external guidance, the Proposer naturally gravitates toward generating intermediate-difficulty questions that require more internal knowledge for reason.

**Dynamics of the sampling efficiency.** As shown in Figure 6a (with external knowledge) and Figure 6b (without external knowledge), the sampling efficiency of the Proposer improves over the course of training in both settings. With external knowledge, the sampling efficiency of the Proposer increases from roughly 15% to over 35%. Without external knowledge, it rises more modestly, from about 15% to approximately 21% as training progresses. This suggests that the Proposer gradually learns to better follow the format instruction and effectively leverage both the external knowledge and/or its internal knowledge to produce QA pairs. Moreover, the Proposer achieves a higher sampling efficiency when external knowledge is provided. This is expected, as the external knowledge offers additional guidance that helps the Proposer generate valid QA pairs more reliably.

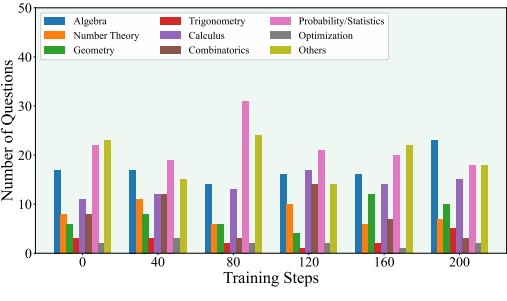 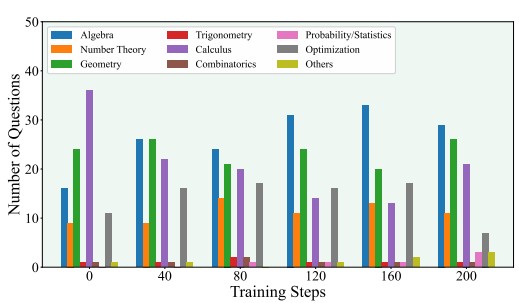

(a) Question diversity distribution w/ $\mathcal{K}$.  (b) Question diversity distribution w/o $\mathcal{K}$.

Figure 8: Analysis on question diversity distribution.

**Dynamics of question diversity distribution.** We use the same categories and similar prompt (See prompt in Figure 15) when analyzing the knowledge pieces in Table 10, and report the question diversity distribution in Figure 8a (with external knowledge) and Figure 8b (without external knowledge).

As shown in Figure 8a, when external knowledge is provided, questions from the Proposer consistently covers a broad range of math domains throughout training. Overall, the number of questions in most domains does not fluctuate significantly across training steps. Algebra, Probability/Statistics, and Other are the most frequently occurring categories, which is consistent with our earlier observations on the knowledge distributions in Table 10. However, for the Other category, the questions occupy a noticeably smaller proportion compared with the distribution of the external knowledge, as shown in Table 10. One possible explanation is that the Proposer tends to generate math-related questions that fall into the first eight domains, even when the provided knowledge piece belongs to the Other category and does not directly involve mathematical content.

When external knowledge is not provided, as shown in Figure 8b, the domain distribution becomes more concentrated. In particular, Algebra, Geometry, and Calculus dominate across all checkpoints. Number Theory and Optimization also appear frequently. This is reasonable because these domains are more strongly represented in the model's internal knowledge and are structurally easier for the model to remix into new question formats. In contrast, domains such as Combinatorics, Trigonometry, and Probability/Statistics occur far less frequently. The proportion of Calculus questions decreases as training progresses, while the proportion of Algebra questions increases. This likely indicates that the Proposer begins to generate a somewhat more diverse range of topics, though the questions tend to be easier overall, according to our earlier analysis. Additionally, questions under the Other category are much rarer compared to the setting with external knowledge, further indicating that external knowledge is the primary driver of out-of-domain question generation. Overall, the absence of external knowledge leads to a notable collapse in question diversity, with the Proposer gravitating toward a narrower set of mathematically simpler topics.

| Method | AIME24 | AIME25 | AMC | GSM8K | MATH | OlympiadBench | AVG |
|---|---|---|---|---|---|---|---|
| Solver | **7.22** | **7.22** | **40.83** | **84.98** | **68.50** | **28.79** | **39.59** |
| Proposer | 3.33 | 4.44 | 22.92 | 62.79 | 51.93 | 21.56 | 27.82 |
| Proposer w/o $\mathcal{K}$ | 3.89 | 5.56 | 30.42 | 65.07 | 55.67 | 24.00 | 30.77 |

Table 14: Results of the Proposer on math benchmarks.

## H.5 USING THE PROPOSER AS THE SOLVER

We further evaluate whether the Proposer alone is capable of solving math problems effectively, given that PasoDoble relies on the Proposer to produce correct answers with the help of external knowledge. The results are reported in Table 14. Compared to the Solver, a Proposer trained with external knowledge performs significantly worse, suggesting that it depends on the provided knowledge rather than its own internal reasoning to obtain correct answers. Interestingly, the Proposer trained without external knowledge also underperforms the Solver. We attribute this gap to a prompting-format mismatch: the Proposer is not trained under the same solution-formatting conventions as the Solver, leading to template inconsistency and degraded performance.

## I  MANUAL INSPECTION ON THE KNOWLEDGE, QUESTION, AND ANSWER ASSOICAITON

During an earlier iteration we manually annotated a few examples to check the relationship between knowledge, question, and answers. See examples in Figure 9 and Figure 10.

**Knowledge type analysis.** We manually inspect the first 50 knowledge pieces from our main analysis in 5.3 where checkpoint 40 in the Qwen3-1.7B offline training setting can generate a question and answer. The results are shown in Table 15. Compared with GPT-5 annotations, human annotations have a greater tendency to categorize knowledge pieces into "Problem" and "Concept/Theorem (Incomplete)", but not for other categories. We leave more investigation to future work.

| Type | Percentage |
|---|---|
| Concept/Theorem | 62.00 |
|   Complete | 36.00 |
|   Incomplete | 34.00 |
| Problem | 50.00 |
| Method | 44.00 |
| Other | 6.00 |

Table 15: Knowledge types

**Knowledge question association.** We take the first 20 knowledge pieces in the dataset in § 5.3 where Proposer checkpoint 40 can generate a question and answer. We feed them into checkpoint 40/80/120/160/200. This yields 100 output in total from the Proposer. Among these, 87 contain QA pairs, and 60 of those 87 have correct answers that match those produced by GPT-5-mini. Due to limited number of data points, we do not sperarely analyze the distribution for each checkpoint as we do in § 5.3. Instead we aggregate the data points together.

| Source | All | Q w/ Corr Ans |
|---|---|---|
| External | 21.84 | 21.67 |
| Internal | 14.94 | 18.33 |
| Both | 63.22 | 60.00 |

Table 16: Question source distributions.

We first examine whether the knowledge piece and question are on the same topic on a high level, and we find that 100% of the 87 questions are on the same topic as their knowledge pieces.

We then perform our source analysis on both the full set of questions (the "All" column in Table 16) and the subset of questions with verified correct answers (the "Q w/ Corr Ans" column in Table 16). We find that human annotations tend to categorize questions into "Internal" and "Both" more, and "External" less. Despite the difference, our conclusions from Table 11 still holds.

| Type | All | Q w/ Corr Ans |
|---|---|---|
| Derivable | 55.17 | 70.00 |
| Not derivable | 44.83 | 30.00 |

Table 17: Derivability of the answers.

**Knowledge answer association**. We again use the 87 QA pairs for analysis. Human annotations suggest that more than half of all answers are derivable solely from external knowledge alone. This closely matches what the annotations from GPT-5-mini suggests.

Among questions with correct answers, the proportion of derivable answers increases to approximately 70%. This indicate that external knowledge provides meaningful guidance for the Proposer, enabling it to derive correct answers for many of the questions it generates.

---

**Knowledge/Question/Answer Analysis Example 1**

**Knowledge:**
The vector space $V$ is defined as $V = \{ p(x) \in P_2 : p(0) = 0 \}$, where $P_2$ denotes polynomials of degree less than or equal to 2. Let $p(x) = a + bx + cx^2$. Since $p(0) = 0$, we get $a = 0$. Thus, any $p(x) \in V$ has no constant term and can be written as $p(x) = bx + cx^2$. Therefore, $\dim(V) = 2$, and a basis for $V$ is $\{x, x^2\}$.

**Knowledge Type:**
Concept/Theorem (Complete); Problem; Method

**Checkpoint 0**
**Question:** Let $V$ be the vector space of polynomials in $P_2$ (polynomials of degree $\leq 2$) such that $p(0) = 0$. Determine the dimension of $V$ and provide a basis for $V$.
**Answer:** 2;   **Ground Truth:** 2.
**Answer derivable from knowledge:** True;   **Question Source:** External.

**Checkpoint 40**
**Question:** Let $V = \{ p(x) \in P_2 : p(0) = 0 \}$, where $P_2$ denotes polynomials of degree $\leq 2$. Determine the dimension of $V$ and provide a basis for $V$.
**Answer:** 2;   **Ground Truth:** 2.
**Answer derivable from knowledge:** True;   **Question Source:** External.

**Checkpoint 80**
**Question:** Find the dimension of the vector space $V = \{ p(x) \in P_2 \mid p(0) = 0 \text{ and } p(1) = 0 \}$, where $P_2$ denotes polynomials of degree $\leq 2$.
**Answer:** 1;   **Ground Truth:** 1.
**Answer derivable from knowledge:** True;   **Question Source:** Both.

**Checkpoint 120**
**Question:** What is the dimension of the space of polynomials in $P_2$ satisfying $p(0) = 0$, $p(1) = 0$, and $p(2) = 0$?
**Answer:** 0;   **Ground Truth:** 0.
**Answer derivable from knowledge:** True;   **Question Source:** Both.

**Checkpoint 160**
**Question:** Let $T : P_2 \to P_2$ be the linear transformation defined by $T(p(x)) = p(x) - p(0)x$. What is the dimension of the kernel of $T$?
**Answer:** 1;   **Ground Truth:** 0.
**Answer derivable from knowledge:** False;   **Question Source:** Both.

**Checkpoint 200**
**Question:** Let $V$ be the vector space of polynomials in $P_2$ (degree $\leq 2$) satisfying $p(0) = 0$. How many polynomials in $V$ satisfy $p(1) = 0$?
**Answer:** 1;   **Ground Truth:** Infinitely many (the 1-dimensional subspace $\{ ax(1 - x) : a \in \mathbb{R} \}$).
**Answer derivable from knowledge:** False;   **Question Source:** Both.

Figure 9: An annotated example for Knowledge/Question/Answer analysis. Ground Truth Answer is generated by GPT5-mini; Answer derivable from knowledge: whether the answer is derivable using the external knowledge alone; Question Source: Whether the question is composed with external knowledge, parametric knowledge, or both.

---

### Knowledge/Question/Answer Analysis Example 2

**Knowledge:**
Homework 2 Solution. For the function $f(x) = 6x^2 + \sin(x)$, perform the following tasks:
(a) Using the first forward finite divided difference method, compute the approximation of the first derivative of the function at $x_i = 1$ with step sizes $h = 0.1$ and $h = 0.05$. (b) Find the true value of $f'(1)$ analytically. (c) Compute the true fractional relative errors for both cases ($h = 0.1$ and $h = 0.05$) using the analytical solution from (b). (d) Using the second forward finite divided difference method, compute the approximation of the second derivative of the function at $x_i = 1$ with step sizes $h = 0.1$ and $h = 0.05$. (e) Find the true value of $f''(1)$ analytically. (f) Compute the true fractional relative errors for both cases ($h = 0.1$ and $h = 0.05$) using the analytical solution from (e).

**Knowledge Type:** Problem.

**Checkpoint 0**
*Omitted (no valid question).*

**Checkpoint 40**
**Question:** Compute the second derivative of $f(x) = 6x^2 + \sin(x)$ at $x = 1$, and provide the exact value.
**Answer:** $12 - \sin(1)$;    **Ground Truth:** $12 - \sin(1)$.
**Answer derivable from knowledge:** False;    **Question Source:** Both.

**Checkpoint 80**
**Question:** Compute the true fractional relative error for the first derivative of $f(x) = 6x^2 + \sin(x)$ at $x = 1$ using a forward finite divided difference method with $h = 0.1$. The true value is $f'(1) = 12 + \cos(1)$.
**Answer:** $1.29\%$;    **Ground Truth:** $4.442\%$ (approximately).
**Answer derivable from knowledge:** False;    **Question Source:** Both.

**Checkpoint 120**
**Question:** Given $f(x) = 6x^2 + \sin(x)$, compute the true fractional relative error for the first derivative at $x = 1$ using the forward finite divided difference method with step size $h = 0.1$. The true derivative is $f'(1) = 12 + \cos(1) \approx 12.5403$.
**Answer:** $4.29\%$;    **Ground Truth:** True fractional relative error $\approx 0.0444217 \approx 4.44217\%$.
**Answer derivable from knowledge:** False;    **Question Source:** Both.

**Checkpoint 160**
**Question:** For $f(x) = 6x^2 + \sin(x)$, compute the true fractional relative error for the first derivative at $x = 1$ using the forward finite divided difference method with $h = 0.1$. The fractional relative error is $\frac{\text{approx} - \text{true}}{\text{true}} \times 100\%$.
**Answer:** $4.44\%$;    **Ground Truth:** $4.44\%$.
**Answer derivable from knowledge:** False;    **Question Source:** Both.

**Checkpoint 200**
**Question:** Compute the first derivative of $f(x) = 6x^2 + \sin(x)$ at $x = 1$ using the forward finite divided difference method with $h = 0.1$ and $h = 0.05$. Also compute the true derivative analytically and the true fractional relative errors.
**Answer:** $13.081$;    **Ground Truth:** $12.54$
**Answer derivable from knowledge:** False;    **Question Source:** External.

Figure 10: An annotated example for Knowledge/Question/Answer analysis. Ground Truth Answer is generated by GPT5-mini; Answer derivable from knowledge: whether the answer is derivable using the external knowledge alone; Question Source: Whether the question is composed with external knowledge, parametric knowledge, or both.

# J  PROMPTS

The Proposer prompt is shown in Figure 11, and the Solver prompt in Figure 12. For analysis of the Proposer in § 5.3, we show the difficulty level evaluation prompt in Figure 13, the knowledge type prompt in Figure 14, the knowledge/question domain prompt in Figure 15, the knowledge-question association prompt in Figure 16, and the knowledge-answer association prompt in Figure 17.

---

### Proposer Prompt

**System prompt**:
You are the proposer in a proposer-solver game. Your task is to create a challenging, well-structured, diverse, and unambiguous mathematical problem that has a verifiable numerical answer, using the provided external and internal knowledge as context.

Enclose the problem statement within <problem>... </problem> tags. Provide a detailed step-by-step solution, including a brief verification or sanity check, within <answer>...</answer> tags. The final numerical result must be enclosed in \boxed{} inside the <answer> section.

**User prompt:**
External knowledge: {{Knowledge}}

Now, please create a challenging, well-structured, diverse, and unambiguous mathematical problem that has a verifiable numerical answer, using the provided external and internal knowledge as context.

---

Figure 11: Prompt for the Proposer.

---

### Solver Prompt

**System prompt**:
Please reason step by step, and put your final answer within \boxed{}.

**User prompt:**
{{Question}}

---

Figure 12: Prompt for the Solver.

---

### Difficulty Level Evaluation Prompt

**System prompt**:
You are a math-question difficulty evaluator. Your task is to assign a difficulty LEVEL $\in \{1, 2, 3, 4, 5\}$ to a target problem, calibrated using the exemplar questions below. Do NOT solve the problem. Only assess its difficulty.

# Difficulty Signals (use qualitatively, not to compute answer):
- Knowledge load: prerequisite theorems/definitions and their sophistication.
- Structural complexity: number of steps, interdependence, casework.
- Strategic demand: need for key insight, representation changes, or non-obvious strategy.
- Abstraction level: modeling, generalization, proof-like reasoning.
- Trickiness: misleading cues or pitfalls.
- Time demand: minutes required for a well-prepared student without tools.

# Procedure
1. Calibrate: observe the 5 exemplar levels and internalize what distinguishes each.
2. Analyze the target question only on difficulty signals (do not compute its answer).
3. Decide difficulty level $\in \{1, 2, 3, 4, 5\}$. If between levels, choose the higher and mention uncertainty.
4. Output structured JSON only.

# Output format (strict JSON, no extra text):
{
   "level": 1|2|3|4|5,
   "justification_short": "60 words; concise, no solution steps",
}

# Examples
*Omitted here*

**User prompt:**
{{Question}}

Now, please evaluate the difficulty of the provided math question.

Figure 13: Prompt for the difficulty level evaluation.

---

### Knowledge Type Evaluation Prompt

**System prompt**:
You are a math-knowledge type evaluator. Your task is to assign a content type to a target knowledge.

# Content type
- concept/theorem: The knowledge describes a mathematical concept (e.g., the definition of a topic or notion) or a theorem (e.g., a statement about the properties of a mathematical object). Because many knowledge pieces are extracted excerpts, their descriptions may be incomplete, for example, presented only in natural language without formal notation or missing key details. We therefore further divide this category into complete (denoted as concept/theorem (complete)) and incomplete (denoted as concept/theorem (incomplete)) subtypes. A knowledge piece that describes one concept/theorem completely but another incompletely is counted in both subcategories.
- problem: The knowledge explicitly describes a mathematical problem to be solved. In many cases, such knowledge pieces are excerpts from homework assignments or problem sets. Some also originate from textbooks or Wikipedia entries describing conjectures or open problems.
- method: The knowledge describes a procedure/technique to solve a type/instantiation of a mathematical problem. In many cases, this type of knowledge also mentions the corresponding problem.
- other: Knowledge pieces that do not fall under any of the above categories (concept/theorem, problem, or method) are classified as none.
A knowledge piece can be categorized into multiple content types if it contain multiple types of content (e.g., it contains both a problem and a method). However, type "other" should be in mutual exclusion with other content types.

# Output format (strict JSON, no extra text):
{
    "content_type": [type_1, ..., type_N],
    "content_type_justification": "Your justification for choosing the content type(s)",
}

# Examples
*Omitted here*

**User prompt:**
{{Knowledge}}

Now, please evaluate the content type for the provided knowledge peice.

---

Figure 14: Prompt for the knowledge type evaluation.

---

### Knowledge/Question Domain Evaluation Prompt

**System prompt**:
You are a knowledge/question-domain evaluator. Your task is to assign a math domain to a target knowledge piece/question.

# Math domain
For math domain, there are 9 domains. They include algebra, number theory, geometry, trigonometry, calculus, combinatorics, probability/statistics, optimization, and others.
A knowledge piece/question can be categorized into ONLY one math domain. However, domain "others" should be in mutual exclusion with other math domains.

# Output format (strict JSON, no extra text):
{
  "math_domain": "Your choice of the math domain (lowercase)",
  "math_domain_justification": "Your justification for choosing the math domain",
}

# Examples
*Omitted here*

**User prompt:**
{{Knowledge/Question}}

Now, please evaluate the math domain for the provided knowledge peice/question.

---

Figure 15: Prompt for the knowledge/question domain evaluation.

---

### Knowledge-question Association Evaluation Prompt

**System prompt**:
You are a knowledge-question association evaluator. Your task is to determine the association type between the external knowledge and the question.

# Knowledge-question association type
- external: The question and its conditions are directly extracted (i.e., copied) from the external knowledge, assuming the external knowledge already contains a question and provides all necessary conditions.
- internal: Neither the question nor its conditions are explicitly stated or implicitly derivable from the external knowledge. In this case, the external knowledge only guides the general topic, while the question is generated entirely from the model's internal parametric knowledge.
- both: The question requires both external and internal knowledge. That is, part of the question and its conditions are explicitly stated or implicitly derivable from the external knowledge, while the remaining parts are not. This includes cases where the question is a rephrasing or modification of a problem described in the external knowledge (e.g., conditions are added, removed, or altered), requiring the model to draw on its internal knowledge. It also includes cases where the question is an instantiation of a concept or problem mentioned in the external knowledge, where the model must interpret that knowledge and supplement it with internally stored information to construct coherent conditions and a complete question.

# Output format (strict JSON, no extra text):
{
    "knowledge_question_association": "external"|"internal"|"both",
    "knowledge_question_association_justification": "Your justification for choosing the math domain.",
}

# Examples
*Omitted here*

**User prompt:**
External Knowledge: {{Knowledge}}

Question: {{Question}}

Now, please evaluate the knowledge-question association for the provided knowledge piece and the question.

Figure 16: rompt for the knowledge-question association evaluation.

---

**Knowledge-answer Association Evaluation Prompt**

**System prompt**:
You are a knowledge–answer association evaluator. Your task is to determine whether the answer is derivable from the provided external knowledge, given the question.

# Knowledge-answer association type
- derivable: The answer can be obtained solely from the provided external knowledge. This includes cases where the relevant facts or conditions are explicitly present in the external knowledge, and any additional reasoning required involves only minimal common-sense steps (e.g., basic arithmetic).
- not derivable: The answer cannot be obtained from the external knowledge alone. This includes cases where the answer requires information not mentioned in the external knowledge, or where the question introduces new conditions or assumptions that are absent from the external knowledge, such that additional parametric or outside knowledge is necessary.

# Output format (strict JSON, no extra text):
{
    "answer_derivable_solely_from_knowledge": true|false,
    "answer_derivable_solely_from_knowledge_justification": "Your justification for claiming whether the answer is derivable solely from the knowledge",
}

# Examples
*Omitted here*

**User prompt:**
External Knowledge: {{Knowledge}}

Question: {{Question}}

Answer: {{Answer}}

Now, please evaluate the knowledge–answer association for the given question, external knowledge, and answer.

---

Figure 17: Prompt for the knowledge-answer association evaluation.

