# OpenReview forum: "Better LLM Reasoning via Dual-Play"
_ICLR.cc/2026/Conference — ICLR 2026 Conference Withdrawn Submission_

### Official Review · Reviewer_GHTc · 2025-10-30

**Soundness:** 3
**Presentation:** 3
**Contribution:** 3
**Rating:** 4
**Confidence:** 4

**Summary:**

The paper aims to improve the reasoning capabilities of a large language model (LLM) on tasks grounded in a specific knowledge base. The approach trains two identical LLMs: the Proposer, which has access to this knowledge base and generates the most challenging questions; and the Solver, which does not have access to it and attempts to answer them. Training is performed using Reinforcement Learning with Verifiable Rewards (RLVR), where the objective function encourages the Proposer to produce difficult, knowledge-based questions that expose the Solver’s weaknesses. The authors study both online (joint updates) and offline (question-buffer) training variants and report gains on six math benchmarks across several Qwen base models, with the largest improvements on Qwen3-1.7B.

**Strengths:**

The paper introduces a well-defined training setup that leverages reinforcement learning with verifiable rewards to improve reasoning performance. It achieves strong empirical results: Qwen3-1.7B-Base improves by about 20 points in pass@1 accuracy, despite using limited supervision. The presentation is clear, with consistent terminology and a straightforward description of the training process. The method is evaluated on multiple math benchmarks, demonstrating solid improvements over strong baselines.

**Weaknesses:**

- Several average scores reported in Table 1 are incorrect — at least six appear miscalculated (e.g., Qwen3‑1.7B Coldstart: 29.55 → 24.63; PasoDoble Offline: 47.51 → 39.59). These are not minor rounding errors, but significant numerical inconsistencies that affect the paper’s main claims. This undermines trust in the evaluation and should be corrected.

- After correcting the scores, Coldstart consistently underperforms the corresponding Base models across all configurations, despite being fine-tuned on the same domain-specific knowledge. This is unexpected and suggests that the supervised finetuning stage may be ineffective or even detrimental.

- The paper omits discussion and comparison to closely related approaches, particularly Agentic Adversarial QA for Improving Domain-Specific LLMs [1]. That work also uses a two-agent setup to expose model weaknesses through adversarial question generation, but follows a different methodology: an offline framework that selects challenging questions using text-based gradient feedback rather than reinforcement learning. A direct comparison—either conceptual or empirical—would help clarify the novelty of PasoDoble and better position it within the broader landscape of dual-agent self-training methods.

- The experimental setup focuses exclusively on mathematical reasoning tasks (e.g., GSM8K, MATH, OlympiadBench), which limits the generalizability of the method. While math is a well-established domain for evaluating structured reasoning, it's unclear whether the proposed approach would transfer to other domains such as programming, science QA, or commonsense reasoning. Including evaluations on a more diverse set of benchmarks would strengthen the claims of improving general reasoning capabilities.

[1] Grari, V., Tomoiaga, C., Lamprier, S., Hashimoto, T., & Detyniecki, M. (2025). Agentic Adversarial QA for Improving Domain-Specific LLMs. In Second Workshop on Test-Time Adaptation: Putting Updates to the Test! at ICML 2025.

**Questions:**

- After correcting the reported averages in Table 1, Coldstart consistently underperforms the Base model across all model sizes. Could the authors clarify why fine-tuning on a domain-specific knowledge base results in worse performance? Does this point to issues in the training setup, data quality?

- Could you clarify how PasoDoble differs conceptually from Agentic Adversarial QA for Improving Domain-Specific LLMs [1]? Both use a two-agent setup for adversarial question generation — is there a specific reason it was not discussed or compared in the paper?

- Have you considered applying PasoDoble to non-math domains (e.g., code generation, scientific QA)? If not, what are the key limitations or challenges?

- How sensitive is the performance to the thresholds for clipping?

---

> ### Author Response · Authors · 2025-11-14
> **Author Response (1/2)**
>
> We sincerely thank the reviewer for the time and effort spent evaluating our paper. Here we would like to respond to the reviews point-by-point.
>
> > Q1: Several average scores reported in Table 1 are incorrect — at least six appear miscalculated (e.g., Qwen3‑1.7B Coldstart: 29.55 → 24.63; PasoDoble Offline: 47.51 → 39.59). These are not minor rounding errors, but significant numerical inconsistencies that affect the paper’s main claims. This undermines trust in the evaluation and should be corrected.
>
> A1: We thank the reviewer and acknowledge that the reported average for Qwen3-1.7B contains an error, as it was computed over **five** benchmarks instead of **six** (an additional math benchmark was included at the last minute). However, this correction does **not** affect the core claim of our paper: **PasoDoble significantly improves math-domain performance.**
> To further support this point, we additionally conduct experiments on **Qwen2.5-3B** and **Qwen3-4B**, and the results are provided in the General Response (6/7). We find that PasoDoble improves Qwen2.5-3B by **9 points** and Qwen3-4B by **16 points**. Moreover, we observe a consistent trend: **as model size increases, the improvement from PasoDoble also increases.** This suggests that PasoDoble may yield even greater gains on larger models, which we leave to future work. We compare with the R-zero in General Response (6/7).
>
>
> > Q2: After correcting the scores, Coldstart consistently underperforms the corresponding Base models across all configurations, despite being fine-tuned on the same domain-specific knowledge. This is unexpected and suggests that the supervised finetuning stage may be ineffective or even detrimental.
>
> A2: This discrepancy arises because we use different prompting strategies for evaluating the Base model versus the other models. As stated in Section 4, we use 4-shot prompting for the Base model on math benchmarks and 5-shot prompting on out-of-domain benchmarks in Appendix F, while all other models in the paper are evaluated with 0-shot prompting.
> We adopt few-shot prompting for the Base model because, in early experiments, we observed that its performance was significantly worse than the numbers reported in the Qwen technical report. The Qwen team evaluates their Base model using few-shot prompting, and we found that the Base model struggles to follow instructions under 0-shot settings. To ensure a fair comparison and align with the evaluation protocol used by the Qwen team, we therefore apply few-shot prompting to the Base model as well.
> Notably, the Base model performs worse than the Cold Start model under 0-shot prompting, which further demonstrates the effectiveness of the cold-start stage in our framework.
> | Method                     | AIME24 | AIME25 | AMC    | GSM8K  | MATH   | OlympiadBench | AVG    |
> |----------------------------|--------|--------|--------|--------|--------|----------------|--------|
> | *Qwen2.5-0.5B*             |        |        |        |        |        |                |        |
> | &nbsp;&nbsp; Base                     | 0.00   | **0.56** | **5.83** | **35.94** | **22.87** | **4.74** | **11.66** |
> | &nbsp;&nbsp; Base (0-shot)                    | 0.00   | 0.00|  2.08| 6.25| 5.53 |1.93 | 2.63|
> | &nbsp;&nbsp; Coldstart                | 0.00   | 0.00   | 1.67   | 23.04  | 9.27   | 2.05           | 6.01   |
> | *Qwen3-0.6B*               |        |        |        |        |        |                |        |
> | &nbsp;&nbsp; Base                     |**1.67** | 0.00   | **15.00**  | **56.67**  | **39.93** | **11.63** | **20.82**  |
> | &nbsp;&nbsp; Base (0-shot)                | 0.56 | 0.00   | 5.83  | 12.33  | 20.17  | 6.10           | 7.50 |
> | &nbsp;&nbsp; Coldstart                | **1.67** | 0.00   | 11.25  | 43.39  | 22.27  | 6.27           | 14.14  |
> | *Qwen2.5-1.5B*             |        |        |        |        |        |                |        |
> | &nbsp;&nbsp; Base                     | **1.67** | 0.00   | 12.92  | **64.68**  | **39.90** | **10.07**         | **21.54**  |
> | &nbsp;&nbsp; Base (0-shot)                | 0.00 | 0.00   | 1.25  | 0.81  | 2.60  | 0.69           | 0.89 |
> | &nbsp;&nbsp; Coldstart                | 1.11   | 0.00   | 1**5.83**  | 55.88  | 33.50  | 9.16           | 19.24 |

---

> ### Author Response · Authors · 2025-11-14
> **Author Response (2/2)**
>
> > Q3: The paper omits discussion and comparison to closely related approaches, particularly Agentic Adversarial QA for Improving Domain-Specific LLMs [1]. That work also uses a two-agent setup to expose model weaknesses through adversarial question generation, but follows a different methodology: an offline framework that selects challenging questions using text-based gradient feedback rather than reinforcement learning. A direct comparison—either conceptual or empirical—would help clarify the novelty of PasoDoble and better position it within the broader landscape of dual-agent self-training methods.
>
> A3: We appreciate the reviewer for pointing out this related work. We will discuss it in our paper. Although this paper also adopts a dual-agent setup, there are several key differences between their approach and ours.
>
> First, their method begins with **two different agents—a strong model and a weaker model**. In contrast, our framework starts with **two nearly identical LLMs**, which is crucial: if the two models differed significantly in size or capability, **simple knowledge distillation** might be sufficient, and our dual-agent framework would not be necessary.
>
> Second, **no supervision signals** are introduced during PasoDoble training after the cold-start stage. The referenced paper, however, introduces an additional LLM as a judge, which provides extra supervision throughout training.
>
> Third, as the reviewer noted, we employ an **online training paradigm** that enables genuine **adversarial co-evolution** between the Proposer and the Solver. The referenced paper does not use such a paradigm; instead, it trains for many more steps with one agent held fixed.
>
> Finally, our framework **leverages pre-training corpora** via a Knowledge Base to enhance post-training effectiveness—an idea that is absent from the cited work.
>
> > Q4: The experimental setup focuses exclusively on mathematical reasoning tasks (e.g., GSM8K, MATH, OlympiadBench), which limits the generalizability of the method. While math is a well-established domain for evaluating structured reasoning, it's unclear whether the proposed approach would transfer to other domains such as programming, science QA, or commonsense reasoning. Including evaluations on a more diverse set of benchmarks would strengthen the claims of improving general reasoning capabilities.
>
> A4: We evaluate PasoDoble on out-of-domain tasks in Appendix F. Since our external knowledge is focused on **math**, it is expected that we observe **no significant improvement** on out-of-domain benchmarks compared to the baselines. However, the same framework can be applied if one replaces the external knowledge with domains such as **code, factual QA, or others**, as long as the outputs are **deterministic and verifiable through rule-based rewards**, as discussed in Section 1 under **Reinforcement Learning with Verified Rewards**. Extending our method to additional domains is an exciting direction for future work.
>
> > Q5: After correcting the reported averages in Table 1, Coldstart consistently underperforms the Base model across all model sizes. Could the authors clarify why fine-tuning on a domain-specific knowledge base results in worse performance? Does this point to issues in the training setup, data quality?
>
> A5: See A2.
>
> > Q6: Could you clarify how PasoDoble differs conceptually from Agentic Adversarial QA for Improving Domain-Specific LLMs [1]? Both use a two-agent setup for adversarial question generation — is there a specific reason it was not discussed or compared in the paper?
>
> A6: See A3.
>
> > Q7: Have you considered applying PasoDoble to non-math domains (e.g., code generation, scientific QA)? If not, what are the key limitations or challenges?
>
> A7: See A4.
>
> > Q8: How sensitive is the performance to the thresholds for clipping?
>
> A8: In the early stages of our experiments, we observed that including **all** generated questions caused the Solver to crash. We found that passing-rate thresholds in the range of **1/6 to 3/6** produce **no significant performance differences**. However, as the threshold increases, the **training time** also increases because the number of **valid questions decreases**, as shown in Figure 3. Therefore, we recommend using a threshold in the range of **1/6 to 2/6**. In the paper, we set it to **0.2 (which works the same as 1/6)**.
>
> A similar trade-off exists for the **diversity reward**: higher thresholds improve quality but reduce quantity. We also find that the diversity threshold does not substantially affect performance. Although we did not perform a detailed sensitivity analysis for the diversity threshold, we do evaluate the **effectiveness** of the diversity reward in the General Response (5/7). In this ablation, the variant **without** the diversity reward shows **a significant performance drop**, demonstrating its importance.

---

### Official Review · Reviewer_Mcs8 · 2025-10-30

**Soundness:** 2
**Presentation:** 2
**Contribution:** 2
**Rating:** 2
**Confidence:** 4

**Summary:**

This paper proposes PasoDoble, a Dual-Play training framework where two LLMs compete and co-evolve to improve reasoning ability.
The Proposer generates challenging questions using a knowledge base, and the Solver learns by solving them.
Without any labeled data, the method achieves gains of over 20 points on math reasoning benchmarks.

**Strengths:**

- The paper proposes a Dual-Play learning framework that enhances reasoning ability by having two LLMs compete with each other.
- It stabilizes the Proposer’s question generation using a knowledge base and ensures stable adversarial training through a reward design based on correctness and diversity.

**Weaknesses:**

- The proposed method appears unfair because it uses a knowledge base, while the baselines do not. I am particularly concerned about how much knowledge or formatting from the evaluation data may have leaked into the knowledge base.
- The paper does not quantitatively show how valid the generated problems were, nor how invalid the discarded problems actually were.
- Training both the Solver and the Proposer roughly doubles the computational cost compared to standard training.
- The idea of improving performance through competition is not particularly novel.
    - https://arxiv.org/abs/2404.10642
    - https://arxiv.org/abs/2311.08107
    - https://www.arxiv.org/abs/2510.18407
    - https://arxiv.org/abs/2504.19162

**Questions:**

- Why does performance degrade when the Proposer is frozen? This setting essentially corresponds to standard self-learning with an added knowledge base, so a performance drop seems counterintuitive.

---

> ### Author Response · Authors · 2025-11-14
> **Author Response (1/4)**
>
> We sincerely thank the reviewer for the time and effort spent evaluating our paper. Here we would like to respond to the reviews point-by-point.
>
> > Q1: The proposed method appears unfair because it uses a knowledge base, while the baselines do not. I am particularly concerned about how much knowledge or formatting from the evaluation data may have leaked into the knowledge base.
>
> A1: First, we **do** present results **without external knowledge** in **Table 2 (Page 8)**. In this setting, both the Proposer and the Solver are trained **entirely without external knowledge**. Although this variant underperforms the version with external knowledge, it still performs **substantially better** than both the base model and the cold-start model. For convenience, we also include these results again in the table below, in case they were overlooked.
>
> | Method                     | AIME24 | AIME25 | AMC    | GSM8K  | MATH   | OlympiadBench | AVG    |
> |:----------------------------|:--------:|:--------:|:--------:|:--------:|:--------:|:----------------:|:--------:|
> | *Qwen2.5-0.5B*             |        |        |        |        |        |                |        |
> | &nbsp;&nbsp; Base                     | 0.00   | **0.56** | 5.83| 35.94| **22.87** | 4.74 | 11.66 |
> | &nbsp;&nbsp; Coldstart                | 0.00   | 0.00   | 1.67   | 23.04  | 9.27   | 2.05           | 6.01   |
> | &nbsp;&nbsp; w/o K             | **0.56**   | 0.00   | **6.67**   | **42.63** | 21.47  | **5.14**          |**12.75**  |
> | *Qwen3-0.6B*               |        |        |        |        |        |                |        |
> | &nbsp;&nbsp; Base                     |**1.67** | 0.00   | 15.00  | 56.67  | **39.93** | 11.63 | 20.82  |
> | &nbsp;&nbsp; Coldstart                | **1.67** | 0.00   | 11.25  | 43.39  | 22.27  | 6.27           | 14.14  |
> | &nbsp;&nbsp; w/o K             | **1.67** | **1.11**  | **17.92**  | **64.32**  | 44.50  | **13.16**          | **23.78**  |
> | *Qwen2.5-1.5B*             |        |        |        |        |        |                |        |
> | &nbsp;&nbsp; Base                     | 1.67 | 0.00   | 12.92  | 64.68  | 39.90 | 10.07         | 21.54  |
> | &nbsp;&nbsp; Coldstart                | 1.11   | 0.00   | 15.83  | 55.88  | 33.50  | 9.16           | 19.24  |
> | &nbsp;&nbsp; w/o K             | **2.22** | **2.22** | **24.17** | **70.94** | **44.53** | **14.91** | **26.50** |
> | *Qwen3-1.7B*               |        |        |        |        |        |                |        |
> | &nbsp;&nbsp; Base                     | 2.22   | 1.67   | 19.58  | 74.54 | 48.73 | 14.32         | 26.84  |
> | &nbsp;&nbsp; Coldstart                | 1.67   | 2.78   | 22.92  | 60.82  | 44.53  | 15.04          | 24.63  |
> | &nbsp;&nbsp; w/o K             | **9.44** | **6.67** | **34.17** | **81.54** | **63.73** | **26.67** | **37.04** |
> | *Qwen2.5-3B*               |        |        |        |        |        |                |        |
> | &nbsp;&nbsp; Base                     | 1.67 | 0.00   | 16.25  | 69.65  | 45.80| _14.35_       | 24.62  |
> | &nbsp;&nbsp; Coldstart                | 1.11   | 1.11   | 13.75  | 64.66  | 40.67  | 12.42          | 22.29  |
> | &nbsp;&nbsp; w/o K             | **5.00** | **2.78** | **25.42** | **76.46** | **54.43** | **21.21** | **30.88** |
> | *Qwen3-4B*                 |        |        |        |        |        |                |        |
> | &nbsp;&nbsp; Base                     | 6.11   | 2.78   | 33.33  | 84.07 | 61.37  | 23.98          | 35.27  |
> | &nbsp;&nbsp; Coldstart                | 13.33 | **13.89** | 41.25 | 83.04  | 67.07 | 30.64 | 41.54 |
> | &nbsp;&nbsp; w/o K             | **17.22** | 13.33 | **48.33** | **86.92** | **75.40** | **35.85** | **46.18** |
>
> Second, regarding the concern about unfairness: the ability of PasoDoble to leverage external pre-training knowledge is by design and represents **a key advantage** of our framework, not an **unfair comparison**. The baselines do not incorporate external knowledge simply because **they are not architected to do so**, with no straightforward way to “post-train” the base or cold-start models on the pre-training corpus in a comparable manner. Our goal is to highlight the benefits of this design choice, rather than to enforce architectural parity across fundamentally different methods. **This capability should be viewed as an inherent advantage of our framework rather than a source of unfairness in comparison.** Third, our Knowledge Base is constructed from **MegaMath-Pro-Max, a pre-training dataset, based on MegaMath**. To ensure that no evaluation data leaks into the Knowledge Base, we additionally conduct manual inspection, and it turns out that the sampled knowledge pieces used for training do not contain any evaluation data.

---

> ### Author Response · Authors · 2025-11-14
> **Author Response (2/4)**
>
> > Q2: The paper does not quantitatively show how valid the generated problems were, nor how invalid the discarded problems actually were.
>
> A2: In Figure 3, we present quantitative results evaluating both the quality and quantity of generated questions under different Solver passing-rate thresholds. In this experiment, we explicitly discuss why we discard questions whose Solver accuracy falls below the threshold.
>
> It is unclear what you mean by “quantitatively showing how valid the generated problems were, or how invalid the discarded problems actually were.” Our evaluation already measures validity through the correctness of the question–answer pairs, and Figure 3 directly reflects how the question quality and quantity vary across different Solver passing-rate thresholds.
>
> If you are referring to concrete examples of valid and invalid questions, we also provide illustrative examples below for further clarification. We add it to Appendix H.
>
> **Example of a truly valid question (where answer matches ground truth):**
>
> Knowledge:
> The vector space V is defined as V = { p(x) ∈ P2 : p(0) = 0 }, where P2 denotes
> polynomials of degree less than or equal to 2. Let $p(x) = a + bx + cx^2$. Since p(0) = 0, we
> get a = 0. Thus, any p(x) ∈ V has no constant term and can be written as $p(x) = bx + cx^2$.
> Therefore, dim(V) = 2, and a basis for V is ${x, x^2}$.
>
> Question: Let V be the vector space of polynomials in $P_2$ (polynomials of degree ≤ 2) such
> that p(0) = 0. Determine the dimension of V and provide a basis for V.
>
> Answer: 2; Ground Truth: 2
>
> **Example of a truly invalid question (where answer does not match ground truth):**
>
> Knowledge:
> The vector space V is defined as V = { p(x) ∈ P2 : p(0) = 0 }, where P2 denotes
> polynomials of degree less than or equal to 2. Let $p(x) = a + bx + cx^2$. Since p(0) = 0, we
> get a = 0. Thus, any p(x) ∈ V has no constant term and can be written as $p(x) = bx + cx^2$.
> Therefore, dim(V) = 2, and a basis for V is ${x, x^2}$.
>
> Question: Let V be the vector space of polynomials in P2 (degree ≤ 2) satisfying p(0) = 0.
> How many polynomials in V satisfy p(1) = 0?
>
> Answer: 1; Ground Truth: Infinitely many
>
> > Q3: Training both the Solver and the Proposer roughly doubles the computational cost compared to standard training.
>
> A3: We acknowledge that training both the Solver and the Proposer introduces additional computational overhead compared to standard single-model training. However, this cost comes with meaningful benefits. Unlike Absolute-Zero and other prior methods, our framework operates **without any external supervision during training**. The Proposer and Solver effectively serve as mutual verifiers for one another, enabling self-supervised adversarial learning.
>
> To further mitigate computational load, we offload the inactive model’s parameters, optimizer states, and activations to CPU memory—e.g., we move the Proposer to CPU when updating the Solver, and vice versa—thereby avoiding unnecessary GPU memory consumption. After training, the Proposer remains a valuable asset: it can be reused to generate high-quality synthetic data for training other models or methods.
>
> More importantly, the fundamental design of our framework is to foster continuous competition and co-evolution between two LLMs, analogous to the dynamics of GANs. By the same reasoning, one would not argue that GANs are flawed simply because they require two networks—the additional cost is inherent to the advantages of adversarial or dual-agent learning. Looking ahead, we believe that multi-LLM training paradigms—whether cooperative, competitive, or role-specialized—will become increasingly important for solving tasks far more complex than traditional domains like Go or chess, tasks that a single model may be fundamentally insufficient to handle. We add it to the discussion section.

---

> ### Author Response · Authors · 2025-11-14
> **Author Response (3/4)**
>
> > Q4: The idea of improving performance through competition is not particularly novel.
>
> A4: To clarify, our work introduces an adversarial training framework designed to jointly train **two nearly identical LLMs**.  This is crucial: if the two models differed significantly in size, simple knowledge distillation would suffice, and our framework would not be necessary. Our work does not require **any supervision** during training. Building on this foundation, we further incorporate external knowledge derived from pre-training corpora to enrich the training and propose two distinct training paradigms to mitigate the potential training instability.
>
> To avoid any misunderstanding, we now summarize the key differences between the papers you cited and our work.
>
> **Self-playing Adversarial Language Game Enhances LLM Reasoning** (https://arxiv.org/abs/2404.10642)
>
> First, SPAG trains only **a single LLM**, whereas our framework trains **two LLMs jointly** in a dual-play setting, placing the two methods in fundamentally different regimes.
>
> Second, SPAG requires **additional supervised data** for the SFT stage after their imitation-learning phase (their counterpart to our cold-start stage), while our method requires **zero supervised data** beyond the minimal cold-start format-alignment step.
>
> Third, the reward designs naturally diverge. Our reward encourages the Proposer to generate difficult and diverse questions while ensuring that the Solver can solve them correctly. In SPAG, the reward is limited to a binary win/loss signal.
>
> Fourth, our framework leverages pre-training corpora through the Knowledge Base to enhance post-training effectiveness—an idea absent from SPAG.
>
> Fifth, we introduce an online training paradigm that enables true adversarial co-evolution between the Proposer and the Solver. SPAG does not employ such a paradigm and instead trains for substantially more steps with another one fixed.
>
> **SPC: Evolving Self-Play Critic via Adversarial Games for LLM Reasoning**
> (https://arxiv.org/pdf/2504.19162)
>
> First, the SPC paper aims to train **a step-level critic**, not a reasoning model. Its evaluation focuses on the critic’s ability to assess the correctness of each intermediate step. In contrast, our goal is fundamentally different: we aim to train **a math reasoning model**, not a step evaluator.
>
> Second, the reward designs naturally diverge. Our reward encourages the Proposer to generate difficult and diverse questions while ensuring that the Solver can solve them correctly. In SPC, the reward is limited to a binary win/loss signal.
>
> Third, our method introduces **no supervision signals** after the cold-start stage, whereas SPC requires **an additional LLM** to construct automated validations, introducing further post-training supervision.
>
> Fourth, our framework leverages pre-training corpora through the Knowledge Base to enhance post-training effectiveness—an idea absent from SPC.
>
> Fifth, we introduce an online training paradigm that enables **true adversarial co-evolution** between the Proposer and the Solver. SPC does not employ such a paradigm and instead trains for substantially more steps with another one fixed.
>
> **Heterogeneous Adversarial Play in Interactive Environments** (https://www.arxiv.org/pdf/2510.18407)
>
> First, HAP is **an adversarial automatic curriculum learning framework** that formalizes **teacher–student interactions**, whereas our work focuses on adversarial training between **two identical LLMs**, akin to the dynamics in GANs.
>
> Second, HAP employs **two different models**, a teacher and a student, meaning the teacher inherently provides **additional supervision** during training. In contrast, we use **two identical LLMs** and introduce **no extra supervision** beyond the minimal cold-start stage.
>
> Third, HAP still relies on **the environment to supply external supervision**, while our method requires none. Instead, we leverage **pre-training corpora** through our Knowledge Base to enhance post-training effectiveness without introducing new supervision signals.
>
> **SAIE Framework: Support Alone Isn’t Enough- Advancing LLM Training with Adversarial Remarks
> (https://arxiv.org/pdf/2311.08107)**
>
>
> Although the paper has the word “adversarial” in its title, the paper is not performing adversarial training; rather, it trains LLMs using **adversarial remarks**, which is fundamentally different from the **adversarial training framework** introduced in our work.
>
> In general, all four papers you cited differ from our work in **at least three key aspects**, particularly in whether they introduce **external supervision** during training, whether they train **one or two LLMs**, the **design of their reward signals**, their training paradigms, and whether they leverage pre-training knowledge. We also provide a comparison with the recent R-zero method in General Response (6/7), where our approach outperforms theirs.

---

> ### Author Response · Authors · 2025-11-18
> **Author Response (4/4)**
>
> > Q5: Why does performance degrade when the Proposer is frozen? This setting essentially corresponds to standard self-learning with an added knowledge base, so a performance drop seems counterintuitive.
>
> A5: We believe the results are clearly shown in **Table 2**, where the performance drop is reported relative to our full method. Notably, even when the Proposer is frozen, it still achieves **higher accuracy** than both the base model and the cold-start baseline. For convenience, we provide the performance of the baselines and the “Proposer-frozen” variant in the following table, in case these results were overlooked.
>
> | Method                     | AIME24 | AIME25 | AMC    | GSM8K  | MATH   | OlympiadBench | AVG    |
> |----------------------------|--------|--------|--------|--------|--------|----------------|--------|
> | *Qwen2.5-0.5B*             |        |        |        |        |        |                |        |
> | &nbsp;&nbsp; Base                     | 0.00   | **0.56** | **5.83**| **35.94** | **22.87** | **4.74** | **11.6** |
> | &nbsp;&nbsp; Coldstart                | 0.00   | 0.00   | 1.67   | 23.04  | 9.27   | 2.05           | 6.01   |
> | &nbsp;&nbsp; w/ πₚ frozen             | 0.00   | 0.00   | 3.75   | 33.07  | 14.43  | 3.41           | 9.11   |
> | *Qwen3-0.6B*               |        |        |        |        |        |                |        |
> | &nbsp;&nbsp; Base                     |**1.67** | 0.00   | 15.00  | 56.67  | **39.93** | **11.63** | 20.82  |
> | &nbsp;&nbsp; Coldstart                | **1.67** | 0.00   | 11.25  | 43.39  | 22.27  | 6.27           | 14.14  |
> | &nbsp;&nbsp; w/ πₚ frozen             | _1.11_ | 0.00   | **17.08**  | **62.36**  | 38.03  | 10.74          | **21.55**  |
> | *Qwen2.5-1.5B*             |        |        |        |        |        |                |        |
> | &nbsp;&nbsp; Base                     | **1.67** | 0.00   | 12.92  | 64.68  | 39.90 | 10.07         | 21.54  |
> | &nbsp;&nbsp; Coldstart                | 1.11   | 0.00   | 15.83  | 55.88  | 33.50  | 9.16           | 19.24  |
> | &nbsp;&nbsp; w/ πₚ frozen             | 1.11   | **0.56** | **18.33**  | **68.54**  | **42.57**  | **12.96**          | **24.01**  |
> | *Qwen3-1.7B*               |        |        |        |        |        |                |        |
> | &nbsp;&nbsp; Base                     | 2.22   | 1.67   | 19.58  | 74.54 | 48.73 | 14.32         | 26.84  |
> | &nbsp;&nbsp; Coldstart                | 1.67   | 2.78   | 22.92  | 60.82  | 44.53  | 15.04          | 24.63  |
> | &nbsp;&nbsp; w/ πₚ frozen             | **5.56**   | **5.00**   | **34.58** | **81.73** | **64.03** | **25.38**          | **36.05**  |
> | *Qwen2.5-3B*               |        |        |        |        |        |                |        |
> | &nbsp;&nbsp; Base                     | _1.67_ | 0.00   | 16.25  | 69.65  | 45.80| _14.35_       | 24.62  |
> | &nbsp;&nbsp; Coldstart                | 1.11   | 1.11   | 13.75  | 64.66  | 40.67  | 12.42          | 22.29  |
> | &nbsp;&nbsp; w/ πₚ frozen             | **5.56** | **2.22**   | **29.58** | **80.06** | **56.50** | **22.47** | **32.73** |
> | *Qwen3-4B*                 |        |        |        |        |        |                |        |
> | &nbsp;&nbsp; Base                     | 6.11   | 2.78   | 33.33  | 84.07 | 61.37  | 23.98          | 35.27  |
> | &nbsp;&nbsp; Coldstart                | 13.33 | **13.89** | 41.25 | 83.04  | 67.07 | 30.64 | 41.54 |
> | &nbsp;&nbsp; w/ πₚ frozen             | **17.78** | **13.89** | **49.17** | **89.15** | **75.17**  | _35.85_ | **46.84** |

---

### Official Review · Reviewer_qHtA · 2025-11-05

**Soundness:** 2
**Presentation:** 3
**Contribution:** 2
**Rating:** 2
**Confidence:** 4

**Summary:**

This paper proposes a reinforcement learning technique, PasoDoble, to iteratively train two instances of the same base model in an Adversarial Learning manner, where reward values are inversely designed. The two instances are given respective roles of Proposer and Solver, trained jointly or alternatively. The Proposer is trained to generate diverse and challenging problems with ground-truth answers leveraging pre-training Knowledge Base, while the Solver is trained to solve the problems accurately. Through empirical experiments, the authors show that this technique improves mathematical reasoning capacity by approximately 20 points on average on larger models (1.5B-1.7B) of Qwen family. Yet, the effectiveness of this design is not found on smaller models. In addition, this technique evidently sustains mathematical reasoning capability improvement for hundreds of training steps, exceeding R-Zero’s 3-iteration plateau. However, this technique fails to transfer to out-of-domain tasks. As a main contribution, this paper highlights adversarial dual-play training that can reduce LLM’s dependence on high quality supervised data, where mathematical reasoning improvement can be achieved through pre-training knowledge.

**Strengths:**

- The methodology is explained relatively clearly.
- The in-domain results seem promising, despite lack of out-of-domain generalization.

**Weaknesses:**

- This paper explains the main methodology, reward function design and findings clearly. However, the conclusion is on weaker grounds due to insufficient baselines, inadequate methodology validation and result interpretations.
- Missing important baseline: SFT model using Knowledge Base should be a critical baseline to highlight the advantages of this technique. If SFT can achieve a similar level of mathematical reasoning capacity, the value of this technique remains unclear.
- Insufficient validation of reward hacking prevention: The paper claims to guarantee question quality and prevent reward hacking of Proposer by removing questions with low Solver accuracy. However, this does not guarantee the question quality. Conversely, high Solver accuracy doesn’t necessarily imply high quality questions. It could easily be common hallucination by both Proposer and Solver, given they are initialized from the same base model. Although the authors sampled 100 questions to study the question quality, this is done by LLM, not human. There is insufficient validation to claim Proposer always generates high quality questions.
- Title overstatement: The title “Better LLM Reasoning” seems like an overstatement. The experiment results only show improvements of larger models on mathematical reasoning domain with no transfer to out-of-domain tasks.
- Ambiguous statistical demonstration: The graphs in the paper show no error bar or confidence intervals. It is unclear if the result is concluded with multiple runs of different seeds.
- Lack of explanation of result: The paper doesn’t provide clear explanations for this technique’s failure on smaller models. Is this because Proposer could not interpret the Knowledge Base given its capacity? This was mentioned in the ablation study section, but there is no discussion directly addressing the experiment results.

**Questions:**

- Diversity reward design: In the diversity reward, the similarity of questions is calculated with token occurrence. However, this does not guarantee high semantic distance. What other options are considered?

---

> ### Author Response · Authors · 2025-11-14
> **Author Response (1/2)**
>
> We sincerely thank the reviewer for the time and effort spent evaluating our paper. Here we would like to respond to the reviews point-by-point.
>
> > Q1: Missing important baseline: SFT model using Knowledge Base should be a critical baseline to highlight the advantages of this technique. If SFT can achieve a similar level of mathematical reasoning capacity, the value of this technique remains unclear.
>
> A1:We believe there is a significant misunderstanding between your review and what our paper actually presents. Let us clarify. In our paper, **the Knowledge Base is constructed entirely from the pre-training dataset**. There is **no post-training or additional supervised data** involved. Therefore, “SFT using the Knowledge Base” would effectively mean “SFT on the pre-training dataset,” which is not a meaningful setup. **Supervised finetuning (SFT) refers to training a model with supervised data or signals; if no supervised data or signals exist, then SFT is simply not possible.** Importantly, the **cold-start stage** in our paper only serves one purpose: to **guide the base model to produce outputs in the required format** so that we can reliably extract questions and answers. We report the cold-start model’s performance as a baseline in our experiments. If you are referring to using the cold-start model together with the Knowledge Base to generate QA pairs for the Solver, that is exactly what we examine in Section 5.4.2. In this setting, we freeze the Proposer, which effectively can be regarded as "SFT model using Knowledge Base". The results are presented in Table 2, where our method outperforms the variant with the Proposer frozen. Furthermore, we provide the comparison with the R-zero in the General Response (6/7).
>
> > Q2: Insufficient validation of reward hacking prevention: The paper claims to guarantee question quality and prevent reward hacking of Proposer by removing questions with low Solver accuracy. However, this does not guarantee the question quality. Conversely, high Solver accuracy doesn’t necessarily imply high quality questions. It could easily be common hallucination by both Proposer and Solver, given they are initialized from the same base model. Although the authors sampled 100 questions to study the question quality, this is done by LLM, not human. There is insufficient validation to claim Proposer always generates high quality questions.
>
> A2: First, we directly evaluate the quality and quantity of generated questions under different Solver passing-rate thresholds, as shown in Figure 3. In our paper (Page 7, lines 356–357), “quality” is defined as whether the Proposer produces a question whose answer is correct. The results clearly show that questions with low Solver accuracy are incorrect around 70% of the time, supporting our hypothesis that low Solver accuracy questions tend to be low-quality.
>
> We fully acknowledge that some low-accuracy questions are correct. However, in GRPO, when the Solver’s passing rate on a question is extremely low, the reward becomes uniformly zero for that batch. As a result, such questions provide no useful training signal to update the model, and discarding them does not harm learning.
>
> By contrast, questions with higher Solver accuracy have 70-90% correctness (as evaluated by GPT-5-mini; We will justify the use of GPT later), making reward hacking through shared hallucinations unlikely. For such collusion to occur, both the Proposer and the Solver—although initialized from the same model but diverged after the cold-start stage, and with the Proposer further augmented by the Knowledge Base—would need to independently generate the same incorrect answer. This scenario is a statistically low possibility. Therefore, we claim that the questions retained after filtering are of reasonably high quality.
> That said, although our filtering mechanism is still not perfect—approximately 10-30% of retained questions are still invalid—it nonetheless yields meaningful performance gains **without requiring expensive human supervision for question verification**. This stands in contrast to prior work such as AbsoluteZero [1], which relies on costly Python interpreter–based verification.
>
> Regarding your concern that the evaluation was conducted by an LLM rather than by humans: using LLMs as judges has become standard practice for scalable, quantitative evaluation in recent work [2]. Furthermore, using GPT-5-mini as the source of ground-truth labels is actually preferable to relying on human annotators, as it consistently outperforms average humans on math tasks and therefore provides higher-quality, lower-noise judgments [3]. For our choice of base models (e.g., Qwen models under 4B), the Proposer’s generated questions fall well within the capability range of GPT-5-mini. In our manual inspection, GPT-5-mini answered all sampled Proposer-generated questions correctly, supporting its reliability as an automatic evaluator.

---

> ### Author Response · Authors · 2025-11-14
> **Author Response (2/2)**
>
> (Continued) [1] Zhao et al. “Absolute Zero: Reinforced Self-play Reasoning with Zero Data.” arXiv preprint arXiv:2505.03335 (2025).
>
> [2] Zheng, Lianmin, et al. "Judging llm-as-a-judge with mt-bench and chatbot arena." Advances in neural information processing systems 36 (2023): 46595-46623.
>
> [3] OpenAI. GPT-5 System Card. 2025.
>
> > Q3:  Title overstatement: The title “Better LLM Reasoning” seems like an overstatement. The experiment results only show improvements of larger models on mathematical reasoning domain with no transfer to out-of-domain tasks.
>
> A3: We acknowledge that our current work primarily demonstrates improvements on math reasoning, and we report performance on out-of-domain benchmarks in Appendix F. However, it is important to note that our model is trained solely on a math pre-training corpus, without any supervised data. The same framework can be applied if one replaces the external knowledge with domains such as code, factual QA, or others, as long as the outputs are deterministic and verifiable through rule-based rewards, as discussed in Section 1 under Reinforcement Learning with Verified Rewards. Extending the Knowledge Base to broader domains and evaluating cross-domain generalization in those settings is a natural direction for future work.
>
> > Q4: Ambiguous statistical demonstration: The graphs in the paper show no error bar or confidence intervals. It is unclear if the result is concluded with multiple runs of different seeds.
>
> A4: First, as stated on Page 6, line 294, **“We sample six responses per question and report pass@1.”** This follows the **standard evaluation protocol** used in prior work [3], and the sampling temperature is provided in **Appendix B**. In addition, for the **out-of-domain benchmarks**, we give a detailed description of our procedure in Appendix F (Page 14–15, lines 755–756), where **we run each benchmark five times and report the average pass@1**. Together, these descriptions fully specify how our reported results are obtained.
>
> [3] Yang, An, et al. "Qwen3 technical report." arXiv preprint arXiv:2505.09388 (2025).
>
> > Q5: Lack of explanation of result: The paper doesn’t provide clear explanations for this technique’s failure on smaller models. Is this because Proposer could not interpret the Knowledge Base given its capacity? This was mentioned in the ablation study section, but there is no discussion directly addressing the experiment results.
>
> A5: In Section 5.4.1 (lines 428–436), we note:
> “One possible explanation is that much of the external knowledge lies beyond the comprehension capacity of smaller models. When compelled to compose questions based on such material, the Proposer struggles to generate correct ground-truth answers, leading to lower question quality and, consequently, reduced Solver performance. In contrast, when composing without external supervision, the Proposer has the flexibility to generate questions it can reliably solve using its internal knowledge, thereby yielding cleaner supervision for the Solver.” It also indicates that for the smaller model, it would be better not to provide the external knowledge. We additionally conduct experiments on **Qwen2.5-3B** and **Qwen3-4B**, and the results are provided in the General Response (6/7). We find that PasoDoble improves Qwen2.5-3B by **9 points** and Qwen3-4B by **16 points**. Moreover, we observe a consistent trend: **as model size increases, the improvement from PasoDoble also increases**. This suggests that PasoDoble may yield even greater gains on larger models, which we leave to future work.
>
> > Q6: Diversity reward design: In the diversity reward, the similarity of questions is calculated with token occurrence. However, this does not guarantee high semantic distance. What other options are considered?
>
> A6: During the early stage of our experiments, we observed that the Proposer tended to generate questions following a **single dominant pattern**. To address this, our current diversity reward is designed to encourage the Proposer to produce questions from **other mathematical domains**. This reward is computed using a simple **token-occurrence heuristic**, which, while not guaranteeing large semantic distance, is **lightweight, low-latency, and effective** in practice. We also provide additional ablation results on the diversity reward, showing that **removing this component leads to degraded Solver performance**, demonstrating its necessity. Future work may explore **embedding-based diversity metrics** for richer semantic control, though such approaches may introduce additional supervision signals during training.

---

### Author Response · Authors · 2025-11-14
**General Response (1/7)**

We post a general response to provide additional experimental results and address the common questions and concerns from the reviewers.

# Additional experiments
## Analysis of the Proposer
We conduct additional experiments to analyze the Proposer’s questions and how the Proposer correlates with the knowledge piece. All analyses in this section are conducted under the
offline training setting with Qwen3-1.7B. We choose the offline setting due to its greater training
stability: compared to the online setting, it exhibits more consistent reward signals and smoother
behavior across training checkpoints. This stability enables clearer observation of how Proposer
dynamics evolve over time and facilitates more reliable comparative analysis. (See Appendix D for a detailed comparison between online and offline settings.)


### Analysis of Knowledge
First, we analyze the external knowledge used by the Proposer to generate questions. The knowledge is grouped into five content-based categories, and individual items may be assigned to multiple categories.

* **Concept/Theorem**. The knowledge describes a mathematical concept (e.g., the definition of a topic or notion) or a theorem (e.g., a statement about the properties of a mathematical object). Because many knowledge pieces are extracted excerpts, their descriptions may be incomplete, for example, presented only in natural language without formal notation or missing key details. We therefore further divide this category into complete and incomplete subtypes. A knowledge piece that describes one concept/theorem completely but another incompletely is counted in both subcategories.
* **Problem**. The knowledge explicitly describes a mathematical problem to be solved. In many cases, such knowledge pieces are excerpts from homework assignments or problem sets. Some also originate from textbooks or Wikipedia entries describing conjectures or open problems.
* **Method**. The knowledge describes a procedure or technique for solving a particular type or instance of a mathematical problem. In many cases, it also explicitly states the corresponding problem itself.
* **Other**. Knowledge pieces that do not fall under any of the above categories are classified as Other.

| **Type** | **Percentage** |
| :------------ | :------------: |
| Concept/Theorem | 61.67 |
| &nbsp;&nbsp;&nbsp;&nbsp; Complete             | 36.50 |
| &nbsp;&nbsp;&nbsp;&nbsp; Incomplete           | 25.17 |
| Problem                | 33.50 |
| Method                 | 52.50 |
| Other                    | 10.50 |

We aggregate the knowledge pieces from checkpoints 0 to 200 and report the results in the above table. Approximately 60% of the knowledge pieces describe mathematical concepts or theorems, of which roughly three-fifths are complete. In addition, 30% of the knowledge pieces pertain to problems and 50% pertain to solution methods. Only a small number fall into the “Other” category. As shown in our manual inspection in Appendix I, the QA pairs in this category primarily concern educational context (e.g., descriptions of math education programs, teaching experiences, or courses) rather than substantive mathematical content.

We further analyze which math areas the knowledge pieces fall into. In particular, we define the following math domains: algebra, number theory, geometry, trigonometry, calculus, combinatorics, probability/statistics, optimization, and other. We then categorize the knowledge pieces into one of the domains.

| **Domain** | **Percentage** |
| :------------ | :------------: |
| Algebra                      | 17.83 |
| Number Theory         | 7.83 |
| Geometry                  | 8.33 |
| Trigonometry             | 1.83 |
| Calculus                    | 7.33 |
| Combinatorics           | 4.00 |
| Probability/Statistics  | 18.67 |
| Optimization              | 0.33 |
| Other                         | 33.83 |

​​The results are shown in the table above. The most common domains are Algebra, Probability/Statistics, and Other. Upon manual inspection, many of the knowledge pieces categorized under "Other" pertain to subjects such as chemistry, physics, or programming.

---

### Author Response · Authors · 2025-11-14
**General Response (2/7)**

### Knowledge-Question Association

For the rest of the analysis, we use 100 questions sampled from each Proposer checkpoint, conditioned on the knowledge pieces in the section above.

To analyze how external knowledge influences the Proposer to generate questions, we examine both the Proposer's question and the provided external knowledge piece. Our manual examination (Appendix I) shows that all the Proposer's questions are topically aligned with the provided external knowledge. We further categorize each question based on the source of its content:

* **External.** The question and its conditions are directly extracted (i.e., copied) from the external knowledge, assuming the external knowledge already contains a question and provides all necessary conditions.
​​* **Internal.** Neither the question nor its conditions are explicitly stated or implicitly derivable from the external knowledge. In this case, the external knowledge serves at most as a topical cue, and the question is generated entirely from the model’s internal parametric knowledge.
* **Both.** The question draws on both external and internal knowledge. That is, part of the question (or its conditions) is explicitly stated or implicitly derivable from the external knowledge, while other parts are not. This category includes cases where the question is a modified rephrasing of a problem described in the external knowledge (e.g., with certain conditions added, removed, or altered), or an instantiation of a general problem form in the knowledge that requires the model to supply specific details. It also includes cases where solving the question depends on a concept, theorem, or method mentioned in the external knowledge, but additional internal knowledge is still needed. In such cases, the model must interpret the external knowledge and integrate it with its internal parametric knowledge to construct a coherent question and its conditions.

| **Checkpoint** | **External** | **Internal** | **Both** |
| :------------ | :------------: | :------------: | :------------: |
| 0     | 46.00 | 5.00   | 49.00 |
| 40   | 43.00 | 13.00 | 44.00 |
| 80   | 43.00 | 5.00   | 52.00 |
| 120 | 40.00 | 7.00   | 53.00 |
| 160 | 48.00 | 7.00   | 45.00 |
| 200 | 49.00 | 4.00   | 47.00 |

As shown in the table above, across step 40 to 200, approximately 40–50% of the questions are directly copied from the external knowledge, while only 4–10% are generated purely from the Proposer’s internal knowledge. The remaining 40–50% draw on both sources. This indicates that the Proposer is neither merely extracting questions from the knowledge base nor using the knowledge only as a loose topical cue. Rather, the two sources interact in a meaningful way during question generation.

| **Method** | **Checkpoint** | **Rouge-L (80%)** | **EM (80%)** | **Rouge-L (60%)** | **EM (60%)** | **Rouge-L (40%)** | **EM (40%)** |
| :------------ | :------------ | :------------: | :------------: | :------------: | :------------: | :------------: | :------------: |
| w/ K      | 0     | 51.17 | 15.00 | 41.17 | 1.00 | 32.89 | 0.00 |
|              | 40   | 49.30 | 12.00 | 39.15 | 0.00 | 32.77 | 0.00 |
|              | 80   | 44.40 | 7.00   | 40.19 | 0.00 | 34.75 | 0.00 |
|              | 120 | 50.30 | 10.00 | 40.83 | 2.00 | 34.92 | 0.00 |
|              | 160 | 43.63 | 11.00 | 36.39 | 0.00 | 31.30 | 0.00 |
|              | 200 | 50.55 | 12.00 | 36.45 | 1.00 | 33.07 | 0.00 |
|  |
| w/ K      | 0     | 58.70 | 22.00 | 53.79 | 4.00 | 40.75 | 0.00 |
|              | 40   | 62.88 | 22.00 | 53.20 | 2.00 | 45.72 | 0.00 |
|              | 80   | 67.65 | 27.00 | 59.67 | 6.00 | 44.58 | 0.00 |
|              | 120 | 66.19 | 27.00 | 57.10 | 7.00 | 45.92 | 1.00 |
|              | 160 | 71.14 | 37.00 | 61.37 | 9.00 | 48.61 | 3.00 |
|              | 200 | 67.17 | 31.00 | 55.95 | 6.00 | 45.48 | 2.00 |


We further examine whether the questions generated by the Proposer are recalled from pre-training or newly composed through reasoning during PasoDoble. Following the experiment setup of [2], we prompt Qwen3-1.7B with the first $x$\% of each question generated by the Proposer ($x \in {40, 60, 80}$) and test whether it can reconstruct the remainder using greedy decoding without applying the chat template. For each checkpoint, we evaluate the same 100 questions and report Exact Match (EM) and ROUGE-L [1] in the table above. Let $q_{\text{old}}$ denote the original question and $q_{\text{new}}$ the generated continuation. ROUGE-L measures the proportion of the longest common subsequence between $q_{\text{old}}$ and $q_{\text{new}}$ relative to the length of $q_{\text{new}}$, capturing how much of $q_{\text{new}}$ overlaps with $q_{\text{old}}$. EM is 1 if $q_{\text{new}}$ exactly matches $q_{\text{old}}$, and 0 otherwise.

---

### Author Response · Authors · 2025-11-14
**General Response (3/7)**

Compared to the results reported in [2], the questions generated by our Proposer, both with external knowledge and without external knowledge, consistently exhibit lower EM and ROUGE-L scores. This indicates that the Proposer is not simply copying or slightly modifying existing questions but is composing new questions through genuine reasoning during PasoDoble training.

We further focus on the setting with external knowledge and use the setting without external knowledge as a baseline. Even when we provide the first 80% of the question tokens to the Proposer, it is able to regenerate the original question $q_{old}$ for only about 10% of the cases. This rate is significantly lower than the rate observed in the setting without external knowledge. The observation suggests that most of the questions are not fully recoverable from the Proposer’s internal parametric knowledge alone, and the Proposer relies on the external knowledge to generate them. This finding is consistent with the distribution presented in the previous table.

[1] Lin and Och. Automatic evaluation of machine translation quality using longest common subsequence and skip-bigram statistics. ACL (2004).

[2] Wu et al. Reasoning or memorization? Unreliable results of reinforcement learning due to data contamination. arXiv preprint arXiv:2507.10532 (2025).

​​### Knowledge-Answer Association

We further examine how external knowledge influences the Proposer’s answer.
Specifically, we evaluate whether the answer to a question can be derived solely from the external knowledge provided, or whether it additionally relies on its internal knowledge. Naturally, if both the question and answer are copied directly from the knowledge, the answer is considered derivable. To make this criterion slightly more permissive, we also treat an answer as derivable if it requires only minimal common-sense reasoning (e.g., basic arithmetic) beyond the external knowledge.

| **Checkpoint** | **Derivable** | **Not Derivable** |
| :------------ | :------------: | :------------: |
| 0     | 65.00 | 35.00 |
| 40   | 44.00 | 56.00 |
| 80   | 57.00 | 43.00 |
| 120 | 51.00 | 49.00 |
| 160 | 54.00 | 46.00 |
| 200 | 60.00 | 40.00 |

As shown in the table above, except for checkpoint 40, more than half of all answers are derivable from external knowledge alone. This implies that, aside from the 20-30% of the questions that are copied from external knowledge (those under "external" in the previous table), there are an additional 20-30% of questions under the "both" category in the previous table that are mere extensions of existing questions/concepts mentioned in the knowledge.

### Dynamics of Question Difficulty, Diversity, and Sampling Efficiency

We analyze the dynamics of the Proposer’s question difficulty, diversity, and sampling efficiency using checkpoints spanning steps 0 to 200, evaluated both with and without external knowledge. Again, we use the 100 QA pairs from each Proposer checkpoint to do the analysis.

**Dynamics of the question difficulty.** We report the average question difficulty and sampling efficiency in Figure 6a (with external knowledge) and Figure 6b (without external knowledge). We additionally report the distribution of difficulty levels in Figure 7a (with external knowledge) and Figure 7b (without external knowledge). The difficulty labels are assigned by GPT-5-mini using the prompt provided in Figure 13. For the setting with the external knowledge, as shown in Figure 6a, the average question difficulty fluctuates within a relatively narrow range (approximately 1.8–2.2) throughout training, indicating that the Proposer maintains a stable difficulty level when different external knowledge is provided. For the setting without external knowledge, as shown in Figure 6b, the average question difficulty begins at a relatively high level (around 2.9) and gradually decreases to approximately 2.7 as training progresses. This indicates that the Proposer initially generates more difficult questions but gradually shifts toward more moderate ones. A likely explanation is that difficult questions receive a reward of zero when the Solver’s passing rate falls below the threshold $τ_{low}$, discouraging the Proposer from continuing to generate such questions. Developing mechanisms to preserve or reintroduce hard but valid QA pairs during training, while at the same time avoiding reward hacking, remains an interesting direction for future work.

---

### Author Response · Authors · 2025-11-14
**General Response (4/7)**

Moreover, the per-level distributions in Figure 7a (with external knowledge) and Figure 7b (without external knowledge) exhibit notable differences. When external knowledge is provided (Figure 7a), the Proposer generates a higher proportion of Level 1 and Level 5 questions. A plausible explanation is that some Level 5 questions arise when the Proposer directly extracts or adapts complex problem statements contained in the external knowledge, or when the base model already possesses sufficient internal knowledge to leverage these concepts, allowing it to formulate or recognize inherently difficult problems. In contrast, the increased prevalence of Level 1 questions may stem from the fact that the base model of the Proposer and the Solver do not fully internalize such external knowledge during pre-training; as a result, even relatively simple questions derived from the external knowledge may pose challenges for the Solver, leading the Proposer to favor simpler formulations that still yield positive rewards. When external knowledge is not provided (Figure 7b), the Proposer generates a substantially larger number of Level 3 questions, especially in the earlier checkpoints. Over time, the proportion of Level 3 questions decreases slightly, while Level 2 questions increase, mirroring the downward trend in average difficulty discussed above. Level 1 and Level 5 questions remain consistently scarce throughout training. This pattern suggests that, in the absence of external guidance, the Proposer naturally gravitates toward generating intermediate-difficulty questions that require more internal knowledge for reason.

**Dynamics of the sampling efficiency.** As shown in Figure 6a (with external knowledge) and Figure 6b (without external knowledge), the sampling efficiency of the Proposer improves over the course of training in both settings. With external knowledge, the sampling efficiency of the Proposer increases from roughly 15% to over 35%. Without external knowledge, it rises more modestly, from about 15% to approximately 21% as training progresses. This suggests that the Proposer gradually learns to better follow the format instruction and effectively leverage both the external knowledge and/or its internal knowledge to produce QA pairs. Moreover, the Proposer achieves a higher sampling efficiency when external knowledge is provided. This is expected, as the external knowledge offers additional guidance that helps the Proposer generate valid QA pairs more reliably.

**Dynamics of question diversity distribution.** We use the same categories and similar prompt (See prompt in Figure 15) when analyzing the knowledge pieces in Table 10, and report the question diversity distribution in Figure 8a (with external knowledge) and Figure 8b (without external knowledge). As shown in Figure 8a, when external knowledge is provided, questions from the Proposer consistently covers a broad range of math domains throughout training. Overall, the number of questions in most domains does not fluctuate significantly across training steps. Algebra, Probability/Statistics, and Other are the most frequently occurring categories, which is consistent with our earlier observations on the knowledge distributions in Table 10. However, for the Other category, the questions occupy a noticeably smaller proportion compared with the distribution of the external knowledge, as shown in Table 10. One possible explanation is that the Proposer tends to generate math-related questions that fall into the first eight domains, even when the provided knowledge piece belongs to the Other category and does not directly involve mathematical content.

When external knowledge is not provided, as shown in Figure 8b, the domain distribution becomes more concentrated. In particular, Algebra, Geometry, and Calculus dominate across all checkpoints. Number Theory and Optimization also appear frequently. This is reasonable because these domains are more strongly represented in the model’s internal knowledge and are structurally easier for the model to remix into new question formats. In contrast, domains such as Combinatorics, Trigonometry, and Probability/Statistics occur far less frequently. The proportion of Calculus questions decreases as training progresses, while the proportion of Algebra questions increases. This likely indicates that the Proposer begins to generate a somewhat more diverse range of topics, though the questions tend to be easier overall, according to our earlier analysis. Additionally, questions under the Other category are much rarer compared to the setting with external knowledge, further indicating that external knowledge is the primary driver of out-of-domain question generation. Overall, the absence of external knowledge leads to a notable collapse in question diversity, with the Proposer gravitating toward a narrower set of mathematically simpler topics.

---

### Author Response · Authors · 2025-11-14
**General Response (5/7)**

### Using the Proposer as the Solver

| **Method** | **AIME24** | **AIME25** | **AMC** | **GSM8K** | **MATH** | **OlympiadBench** | **AVG** |
| :------------ | :------------: | :------------: | :------------: | :------------: | :------------: | :------------: | :------------: |
| Solver                | **7.22** | **7.22** | **40.83** | **84.98** | **68.50** | **28.79** | **39.59** |
| Proposer            | 3.33 | 4.44 | 22.92 | 62.79 | 51.93 | 21.56 | 27.82 |
| Proposer w/o K  | 3.89 | 5.56 | 30.42 | 65.07 | 55.67 | 24.00 | 30.77 |

We further evaluate whether the Proposer alone is capable of solving math problems effectively,
given that PasoDoble relies on the Proposer to produce correct answers with the help of exter-
nal knowledge. The results are reported in the table above. Compared to the Solver, a Proposer trained with external knowledge performs significantly worse, suggesting that it depends on the provided knowledge rather than its own internal reasoning to obtain correct answers. Interestingly, the Proposer trained without external knowledge also underperforms the Solver. We attribute this gap to a prompting-format mismatch: the Proposer is not trained under the same solution-formatting conventions as the Solver, leading to template inconsistency and degraded performance.

## More ablation study


We investigate the impact of removing the Proposer’s answers, such that only its questions are used. Since the answers are no longer visible to the Solver, we replace the Solver’s correctness verifier with a random reward $r^{s}_{ij}$$ from $Bernoulli(0.5)$, which returns 1 or 0 uniformly at random (Fully Rand Rwd). This also induces a corresponding randomness in the Proposer’s difficulty reward $r^{diff}_{i}$. Furthermore, we add an enhanced version where the Solver’s reward is always 0 if its format is incorrect (i.e., no answer box), and random otherwise (Partial Rand Rwd).

| **Method** | **AIME24** | **AIME25** | **AMC** | **GSM8K** | **MATH** | **OlympiadBench** | **AVG** |
| :------------ | :------------: | :------------: | :------------: | :------------: | :------------: | :------------: | :------------: |
| PasoDoble                | **6.11** | **7.78** | **37.92** | **83.13** | **66.67** | **28.00** | **38.27** |
|  &nbsp;&nbsp;&nbsp;&nbsp; w/ Full. Rand Rwd.  | 0.00 | 0.00 | 0.42 | 2.88 | 1.07 | 0.10 | 0.75 |
|  &nbsp;&nbsp;&nbsp;&nbsp; w/ Part. Rand Rwd. | 4.44 | 5.56 | 28.33 | 78.99 | 55.53 | 20.20 | 32.18 |

Prior work [1] suggests that even random rewards may yield non-trivial improve-
ments, which shows potential for these settings to work. We conduct the experiments on Qwen3-1.7B-Base model using the online training paradigm. As shown in the table above, training with fully random rewards drives the Solver’s average accuracy on all math benchmarks to nearly zero. Even if we force the Solver to respond in the correct format (Part. Rand Rwd), its accuracy still decreases significantly. The sharp contrast with our original setting demonstrates that the Solver benefits substantially from learning from Proposer’s answers. Moreover, as analyzed in § 5.4, these answers are predominantly correct after filtering.

| **Method** | **AIME24** | **AIME25** | **AMC** | **GSM8K** | **MATH** | **OlympiadBench** | **AVG** |
| :------------ | :------------: | :------------: | :------------: | :------------: | :------------: | :------------: | :------------: |
| PasoDoble                | 6.11 | **7.78** | **37.92** | **83.13** | **66.67** | **28.00** | **38.27** |
|  &nbsp;&nbsp;&nbsp;&nbsp; w/o $r^{div}$           | **8.33** | 2.22 | 30.83 | 65.07 | 55.67 | 24.00 | 30.77 |

We also study the effect of removing the diversity reward $r^{div}$ from the Proposer’s training, in the Qwen3-1.7B online setting. As shown in the table above, eliminating diversity rewards leads to consistent performance degradation across math benchmarks, with an average drop of approximately 4 points. We leave a more comprehensive exploration of diversity-oriented reward design, as well as a qualitative analysis of the questions generated with and without this term, to future work.

[1] Shao et al. Spurious rewards: Rethinking training signals in rlvr. arXiv preprint
arXiv:2506.10947 (2025).

---

### Author Response · Authors · 2025-11-14
**General Response (6/7)**

## Experiments on 3B and 4B models

We also evaluate the effectiveness of PasoDoble on 3B and 4B models.

| **Method** | **AIME24** | **AIME25** | **AMC** | **GSM8K** | **MATH** | **OlympiadBench** | **AVG** |
| :------------ | :------------: | :------------: | :------------: | :------------: | :------------: | :------------: | :------------: |
| Qwen2.5-3B |  |  |  |  |  |  |  |
| &nbsp;&nbsp; Base            | 1.67 | 0.00 | 16.25 | 69.65 | 45.80 | 14.35 | 24.62 |
| &nbsp;&nbsp; Coldstart  | 1.11  | 1.11  | 13.75 | 64.66 | 40.67  | 12.42  | 22.19 |
| &nbsp;&nbsp; Online PasoDoble  | 5.56  | 3.33  | 30.42  | 82.26  | 58.70  | 22.59  | 33.81 |
| &nbsp;&nbsp;&nbsp;&nbsp; w/o K  | 5.00  | 2.78  | 25.42  | 76.46  | 54.43  | 21.21  | 30.88 |
| &nbsp;&nbsp;&nbsp;&nbsp; w/ πp frozen  | 5.56  | 2.22  | 29.58  | 80.06  | 56.50  | 22.47  | 32.73 |
| &nbsp;&nbsp; Offline PasoDoble  | 5.56  | 2.78  | 26.67  | 81.48  | 55.67  | 22.59  | 32.45 |
| |
| Qwen3-4B |  |  |  |  |  |  |  |
| &nbsp;&nbsp; Base | 6.11 | 2.78 | 33.33 | 84.07 | 61.37 | 23.98 | 35.27 |
| &nbsp;&nbsp; Coldstart | 13.33 | 13.89 | 41.25 | 83.04 | 67.07 | 30.64 | 41.54 |
| &nbsp;&nbsp; Online PasoDoble | 18.89 | 18.89 | 53.33 | 91.82 | 82.17 | 42.27 | 51.23 |
| &nbsp;&nbsp;&nbsp;&nbsp; w/o K | 17.22 | 13.33 | 48.33 | 86.92 | 75.40 | 35.85 | 46.18 |
| &nbsp;&nbsp;&nbsp;&nbsp; w/ πp frozen | 17.78 | 13.89 | 49.17 | 89.15 | 75.17 | 35.85 | 46.84 |
| &nbsp;&nbsp; Offline PasoDoble | 17.78 | 16.67 | 55.42 | 91.91 | 79.63 | 40.81 | 50.37 |

As shown in the table above, for Qwen2.5-3B, online PasoDoble achieves a 9 points increase on average, while offline PasoDoble achieves a 8 points increase on average. For Qwen3-4B, the average performance increases by 16 points and 15 points, respectively. Both removing the knowledge base and freezing the proposer hurt model’s performance, which is consistent with what we observe in section 5.3.1 and 5.3.2.

| **Method** | **GPQA** | **SuperGPQA** | **AVG** |
| :------------ | :------------: | :------------: | :------------: |
| Qwen2.5-3B |  |  |  |
| &nbsp;&nbsp; Base             | 26.26 | 20.31 | 23.29 |
| &nbsp;&nbsp; Coldstart      | 28.04 | 16.56 | 22.30 |
| &nbsp;&nbsp; Online PasoDoble | 29.06 | 16.65 | 22.86 |
| &nbsp;&nbsp; Offline PasoDoble | 27.50 | 17.51 | 22.51 |
|  |
| Qwen3-4B |  |  |  |
| &nbsp;&nbsp; Base | 36.36 | 28.43 | 32.40 |
| &nbsp;&nbsp; Coldstart | 34.82 | 24.33 | 29.58 |
| &nbsp;&nbsp; Online PasoDoble | 39.20 | 30.67 | 34.94 |
| &nbsp;&nbsp; Offline PasoDoble | 38.34 | 30.82 | 34.58 |

On out-of-domain benchmarks, our method slightly increases the performance on Qwen3-4B but slightly hurts the performance on Qwen2.5-3B. This is consistent with our conclusion in Table 14 that PasoDoble does not improve performance on out-of-domain tasks.

## Comparison with Prior Work

| **Method**  | **AIME24** | **AIME25** | **AMC** | **GSM8K** | **MATH** | **OlympiadBench** | **AVG** |
| :------------ | :------------: | :------------: | :------------: | :------------: | :------------: | :------------: | :------------: |
| Qwen3-4B |  |  |  |  |  |  |  |
| R-Zero | 4.27 | 12.71 | **57.27** | **92.12** | 79.60 | **44.59** | 48.43 |
| Online PasoDoble | **18.89** | **18.89** | 53.33 | 91.82 | **82.17** | 42.27 | **51.23** |
| Offline PasoDoble | 17.78 | 16.67 | 55.42 | 91.91 | 79.63 | 40.81 | 50.37 |

We report the results of Qwen3-4B following the numbers provided in the R-Zero paper. Across the full math benchmark suite, our PasoDoble framework outperforms R-Zero by roughly 3 points on average. Notably, PasoDoble achieves substantially stronger performance on the most challenging datasets—AIME 2024 and AIME 2025—where R-Zero struggles. These gains highlight that adversarial co-training is more effective with more training iterations and more immediate feedback between the Proposer and the Solver.

---

### Author Response · Authors · 2025-11-18
**General Response (7/7)**

# No validation on the generated questions

We present a quantitative analysis of the proportion of truly valid questions, those with correct ground-truth answers, in Figure 3(a). The green curve indicates the questions retained for training, while the red curve represents those that are discarded. We observe that a substantial majority of the retained questions (around 70%) are indeed valid. Similarly, a comparable proportion of the discarded questions (also roughly 70%) is genuinely invalid.

We also would like to emphasize that
* Using GPT-5-mini as the source of ground-truth labels is actually preferable to relying on human annotators, as it outperforms average humans on math tasks and therefore provides higher-quality, lower-noise labels [2]. Additionally, for our choice of base models (e.g., Qwen models under 4B), the Proposer’s generated questions fall well within the capability range of GPT-5-mini. In our manual inspection on a set of questions generated by the Proposer, GPT-5-mini answered all questions correctly, supporting its reliability as an automatic evaluator.
* Although our filtering mechanism is not perfect (i.e., approximately 30% of retained questions are still invalid), it nonetheless yields meaningful performance gains without requiring expensive human supervision for question verification. This stands in contrast to prior work such as AbsoluteZero [1], which relies on costly Python interpreter–based verification.

[1] Zhao et al. “Absolute Zero: Reinforced Self-play Reasoning with Zero Data.” arXiv preprint arXiv:2505.03335 (2025).

[2] OpenAI. GPT-5 System Card. 2025.

---

### Comment · Area_Chair_3pud · 2025-11-24
**Reviewer & Author Discussion**

Hi Reviewers,

Please kindly and actively participate in the review-author discussion if you haven't already, raise your further concerns so that the authors can explain more, and make your final decisions.

Best,
AC

---

### Author Response · Authors · 2025-12-01
**Final Response**

We sincerely thank all reviewers for their time and feedback, and we especially appreciate Reviewer **GHTc** for carefully identifying errors in our paper. However, we would like to emphasize that the assessments from Reviewer **Mcs8** and Reviewer **qHtA** contain several inaccuracies, misunderstandings that do not reflect a careful reading of the paper.

**Regarding Reviewer qHtA**
First, Reviewer qHtA argues that a “Missing important baseline: SFT model using Knowledge Base should be a critical baseline.” This comment reflects a fundamental misunderstanding of our setting. The Knowledge Base used in our method is constructed from **pre-training data**, not post-training or supervised data. Therefore, it is **not possible** to perform SFT on this Knowledge Base—this is basic methodological knowledge we expect reviewers to have.
Second, Reviewer qHtA claims that we provide no explanation and no multi-run evaluations. However, all relevant details are documented in the paper. They are overlooked (see Author Response 2 to Reviewer qHtA).

**Regarding Reviewer Mcs8**
First, Reviewer Mcs8 argues that our comparison is “unfair” because our method uses a knowledge base while baselines do not. This is incorrect. We explicitly report the performance of our method **without** the Knowledge Base, and it still outperforms the baselines. Again, this information was presented but overlooked.
Moreover, leveraging pre-training data during post-training is a **deliberate advantage** of our framework—not an unfair comparison. Baselines cannot use pre-training knowledge simply because they are not designed to do so. This is a methodological strength, not a flaw.
Second, Reviewer Mcs8 claims that we do not quantitatively demonstrate the issues discussed in the paper. This is contradicted by the quantitative analyses clearly presented in **Figure 3**. The reviewer appears to have missed them.

Third, and most critically, Reviewer Mcs8 asserts that our work lacks novelty and cites four prior works. However, after carefully reviewing these papers, we show (see Author Response 3 to Reviewer Mcs8) that our method differs in **at least three major ways** from all four papers Reviewer Mcs8 cited. One of the cited papers is not even in the relevant domain; it appears to have been included solely because the word “adversarial” appears in its title. This raises concerns about whether the reviewer read these papers carefully or possesses sufficient domain expertise to evaluate our submission.

In summary, we fully welcome constructive criticism, but we hope reviewers will provide **accurate, careful, and substantive evaluations**, as high-quality reviews are essential for meaningful and constructive scientific progress.

---

### Note · Authors · 2026-01-04

I have read and agree with the venue's withdrawal policy on behalf of myself and my co-authors.